# Optimal Flow Transport and its entropic regularization: a GPU-friendly matrix iterative algorithm for Flow Balance Satisfaction

**Liangliang Shi**[1], **Yufeng Li**[1], **Kaipeng Zeng**[1], **Yihui Tu**[2], **Junchi Yan**[13*]

[1]Sch. of Computer Science & Sch. of Artificial Intelligence, Shanghai Jiao Tong University
[2]Department of Mathematics, Shanghai University   [3]Shanghai Artificial Intelligence Laboratory
{shiliangliang, liyufeng, zengkaipeng}@sjtu.edu.cn
tuyihui@shu.edu.cn, yanjunchi@sjtu.edu.cn
Code: https://github.com/asimfish/EOFT/

## Abstract

The Sinkhorn algorithm, based on Entropic Regularized Optimal Transport (OT), has garnered significant attention due to its computational efficiency enabled by GPU-friendly matrix-vector multiplications. However, vanilla OT primarily deals with computations between the source and target nodes in a bipartite graph, limiting its practical application in real-world transportation scenarios. In this paper, we introduce the concept of Optimal Flow Transport (OFT) as an extension, where we consider a more general graph setting and the marginal constraints in vanilla OT are replaced by flow balance constraints. To obtain solutions, we incorporate entropic regularization into the OFT and introduce virtual flows for individual nodes to tackle the issue of potentially numerous isolated nodes lacking flow passages. Our proposition, the OFT-Sinkhorn algorithm, utilizes GPU-friendly matrix iterations to maintain flow balance constraints and minimize the objective function, and theoretical results for global convergence is also proposed in this paper. Furthermore, we enhance OFT by introducing capacity constraints on nodes and edges, transforming the OFT problem into a minimum-cost flow problem. We then present the Capacity-Constrained EOFT-Sinkhorn algorithm and compare it with the traditional Minimum cost flow (MCF) algorithm, showing that our algorithm is quite efficient for calculation. In particular, our EOFT-Sinkhorn is evaluated on high-precision and integer-precision MCF problems with different scales from one hundred to five thousand size, exhibiting significant time efficiency and the ability to approximate optimal solutions.

## 1 Introduction

Optimal Transport (OT) has shown increasingly important role in solving various problems in machine learning, including domain adaptation (Tzeng et al., 2017; Cui et al., 2018), generative models (Arjovsky et al., 2017; Li et al., 2022), self-supervised contrastive learning (Caron et al., 2020; Shi et al., 2023), and long-tail recognition (Peng et al., 2021; Shi et al., 2024b) etc. As an efficient method for solving OT, the Sinkhorn algorithm relies on matrix-vector iterations to solve the transportation problem, which is GPU-friendly with high-speed calculations. However, the Sinkhorn algorithm's applicability is limited to bipartite graphs, as it only involves computations between sets of source and target nodes, which significantly deviates from real transportation scenarios. Naturally, a question arises: can optimal transport with matrix iterations be extended to more general graphs?

Numerous researchers have made significant efforts in the field of optimal transport on graphs. For instance, based on the graph structure, (Le et al., 2022) propose a variant of OT called Sobolev transport, which provides a closed-form solution for efficient computation. Additionally, in a related work, (Le et al., 2024) utilize a specific class of convex functions with an Orlicz structure to introduce

---

*Junchi Yan is the correspondence author. This work was in part supported by by NSFC (62222607) and Shanghai Municipal Science and Technology Major Project under Grant 2021SHZDZX0102.

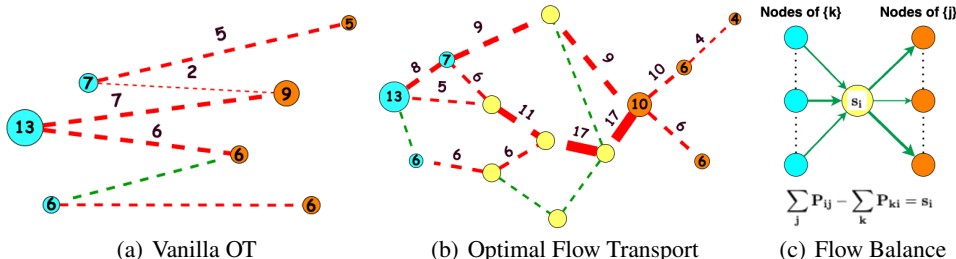

Figure 1: Illustration of our optimal flow transport and difference to vanilla OT. **(a):** Vanilla OT focuses on solving the transportation within a bipartite graph, focusing on transportation between source and target nodes. **(b):** Our OFT aims to solve transportation on a more general graph, focusing on the flow between each pair of nodes. **(c):** In our OFT, we consider the constraints on each node satisfying the flow balance constraint.

the generalized form of Sobolev transport(ST). However, previous works have mostly overlooked the flow balance constraints on nodes in graph transport. They indirectly compute the flow between two nodes by calculating the shortest paths between source and target nodes , thus failing to incorporate capacity constraints on nodes and edges, which are common in real-world transportation scenarios.

In this paper, we aim to fill this gap by proposing Optimal Flow Transport (OFT), in which the constraints for marginals are replaced by flow balance constraints on nodes. As illustrated in Figure 1, we consider more complex transportation problems where the source nodes (blue) may not directly transport to the target nodes (orange). Instead, it may pass through other nodes, such as transshipment points (yellow). As a result, the original transportation problem is generalized into a broader min-cost flow (MCF) problem where the marginal constraints are replaced by flow balance constraints as shown in Figure 1(c). However, in the current works, we are unable to derive a matrix-vector iterative algorithm for flow balance satisfaction in OFT by introducing an entropic regularization (Benamou et al., 2015) directly. This limitation can be attributed to the following reasons: 1) the marginals are unknown in OFT, and we only have information about their differences; 2) the presence of isolated points in the graph that do not carry any flow contradicts the non-sparse nature of matrix iteration.

To overcome these limitations, in this paper, we propose entropic optimal flow transport, in which we do the following reformulation for the flow problem: 1) we introduce virtual transport flows from each node to itself in the graph, allowing isolated points to participate in the iterations even if they do not have any incoming or outgoing flows originally; 2) reformulating the flow balance constraints with marginal-like constraints via adding a new marginal variable for the optimization; 3) Finally adding the entropic regularization to get the relaxation for OFT. The above reformulations of OFT enable us to iteratively update the coupling and marginals to compute an approximate solution for the OFT problem, where we refer to this algorithm as the OFT-Sinkhorn algorithm, and the theoretical guarantee is also proposed for convergence. Moreover, we can impose capacity constraints on nodes and edges in our OFT-Sinkhorn algorithm, in which the proposed method can serve as a matrix iteration-based solver for the minimum-cost flow problem, and experimental results demonstrate the efficiency of our algorithm with minimal computational errors. Finally, this paper contributes:

- We extend the vanilla OT within a bipartite graph to a more general graph case, and thus propose optimal flow transport, in which the marginal constraints are replaced by flow balance constraints.
- We propose the entropic OFT to derive the GPU-friendly OFT-Sinkhorn algorithm to get the approximate solution of the OFT problem. The global convergence is theoretically guaranteed.
- We incorporate node and edge capacity constraints into the OFT, and in this case our OFT is equivalent to the minimum-cost flow problem. By considering these constraints, we modify the OFT-Sinkhorn algorithm to ensure that the output satisfies capacity constraints. Experimental results on the minimum-cost flow problem showcase the superiority of our algorithm.

## 2    RELATED WORKS AND BACKGROUND

**Entropic Optimal Transport.** The Optimal Transport (OT) theory can be traced back to (Monge, 1781) where the objective is to seek a mapping that minimizes the total cost of transporting mass

from a source measure to a target measure, which has various applications in visual matching (Wang et al., 2013), long-tailed learning (Shi et al., 2024b;c), time series analysis (Zhang et al., 2020; Shi et al., 2020), multi-modal learning (Shi et al., 2024a; Wang et al., 2024), and etc. Specifically, given the cost matrix $\mathbf{C}$ and two marginals $(\mathbf{a}, \mathbf{b})$, Kantorovich's OT (Kantorovich, 1960) with entropic regularization involves solving the coupling $\mathbf{P}$ by:

$$\min_{\mathbf{P} \in U(\mathbf{a},\mathbf{b})} < \mathbf{C}, \mathbf{P} > -\epsilon H(\mathbf{P}), \quad \text{where} \quad U(\mathbf{a}, \mathbf{b}) = \{\mathbf{P} \in \mathbb{R}_{mn}^+ | \mathbf{P}\mathbf{1}_n = \mathbf{a}, \mathbf{P}^\top \mathbf{1}_m = \mathbf{b}\} \quad (1)$$

Note that $H(\mathbf{P}) = - < \mathbf{P}, \log \mathbf{P} - \mathbf{1}_{m \times n} >$ represents the entropic regularization, and $\epsilon$ is the coefficient. When $\epsilon = 0$, the entropic OT degenerates to vanilla Kantorovich, and when $\epsilon > 0$, the objective in Eq. 1 is $\epsilon$-strongly convex. Consequently, it possesses a unique solution that can be determined using Sinkhorn algorithms, as discussed in (Cuturi, 2013; Benamou et al., 2015).

**Proposition 1** (Solution Form of Entropic OT (Peyre & Cuturi, 2019)). *The solution to Eq. 1 is unique and satisfy* $\mathbf{P} = diag(\mathbf{u})\mathbf{K}diag(\mathbf{v})$, *where* $\mathbf{K} = e^{-\mathbf{C}/\epsilon}$ *is the Gibbs kernel associated to the cost matrix* $\mathbf{C}$ *and* $(\mathbf{u}, \mathbf{v})$ *are two (unknown) scaling variables.*

With the solution form of entropic OT, Sinkhorn algorithms (Cuturi, 2013; Benamou et al., 2015) were proposed to obtain the solution of Eq. 1, where one can iteratively update $\mathbf{u} = \mathbf{a}/(\mathbf{K}\mathbf{v})$ and $\mathbf{v} = \mathbf{b}/(\mathbf{K}^\top \mathbf{v})$ to approximate the solution of the transportation problem. However, research on solving operational research problems using GPU-friendly matrix iteration algorithms seems to be focused only on transportation problems or matching problems, rather than other complex problems in operation research. In this paper, we propose to use matrix-vector iterative algorithms to solve a more generalized transportation problem, i.e. MCF. To the best of our knowledge, our algorithm is the first GPU-friendly matrix iterative method specifically designed for solving MCF.

**Optimal Transport on the Graph.** The concept of optimal transport on graphs can be attributed to (Feldman & McCann, 2002), who initially compute the shortest distances between source and target nodes to establish a cost matrix. This matrix is then used to calculate the 1-Wasserstein distance, transforming the problem into a linear program, specifically a min-cost flow problem. This methodology has been applied and expanded to formulate and analyze traffic congestion models. Recently, (Le et al., 2022) introduced a new variation named Sobolev transport (ST), tailored for measures supported on graphs, enabling a closed-form expression for quicker computation. Furthermore, (Le et al., 2024) extended Sobolev transport with an Orlicz structure (Orlicz, 1932). They need to first calculate shortest paths before solving the optimal transport between source and target and they can't handle capacity constraints at nodes. In this paper, we introduce optimal flow transport and its entropic regularized case, which calculate the flow in the graph via matrix iterative methods without the need for precomputing the shortest distances on the graph.

**Minimum Cost Flow.** Machine learning has been leveraged to handle optimization problems (Bengio et al., 2021). Minimum-cost flow (MCF) (Goldberg, 1997) is one of the optimization problems from the network flow theory (Iri, 1996), which involves finding the minimum-cost transportation of the specified amount of a commodity from a set of supply nodes to a set of demand nodes in a directed network, considering capacity constraints and linear cost defined on the arcs. The MCF problem arises in diverse applications, including transportation (Ahmady & Eftekhari Yeghaneh, 2022), logistics (Krile, 2004), and many other industrial scenes (Choi et al., 1988). Efficient algorithms have been developed to solve the MCF problem. For instance, Fulkerson's out-of-kilter algorithm (Fulkerson, 1961) takes advantage of the special structure of MCF and adjusts the arcs that do not satisfy optimality properties. The cycle canceling algorithm is proposed by Klein (Klein, 1967), which focuses on maintaining feasible flow at each iteration. Recent years, advancements in the MCF problem have been marked by several pivotal studies. However, most of the previous methods is based on CPU calculations and also make an assumption that all the data is integers (de Vos, 2023), which limits the solution of the MCF problem when dealing with high-precision data. In this paper, we solve MCF by using a GPU-friendly matrix iterative method to get the approximate solutions.

## 3 METHODOLOGY

### 3.1 OPTIMAL FLOW TRANSPORT

In this subsection, We give the definition of Optimal Flow Transport and its exact methods for solving.

**Formulation of OFT** We begin by defining the OFT as follows. Consider $\alpha = \sum_{i=1}^{n} \mathbf{a}_i \delta_{\mathbf{v}_i}$ and $\beta = \sum_{i=1}^{n} \mathbf{b}_i \delta_{\mathbf{v}_i}$ are two measures on the graph $G(\mathcal{V}, \mathcal{E})$, where $N = |\mathcal{V}|$ and $(\mathbf{a}, \mathbf{b})$ are two balanced vectors satisfying $< \mathbf{a} - \mathbf{b}, \mathbf{1} >= 0$. The formulation of OFT can be specified as:

$$\min_{\mathbf{P} \geq \mathbf{0}} < \mathbf{D}, \mathbf{P} > \quad \text{s.t.} \quad \mathbf{P}\mathbf{1}_N - \mathbf{P}^\top \mathbf{1}_N = \mathbf{s} \tag{2}$$

where $\mathbf{D} \geq \mathbf{0}$ is the distance matrix and $\mathbf{s} = \mathbf{a} - \mathbf{b}$. Note for each node $\mathbf{v}_i \in \mathcal{V}$, if $\mathbf{s}_i > 0$, we refer to node $\mathbf{v}_i$ as a supply node with a supply value of $\mathbf{s}_i > 0$. If $\mathbf{s}_i < 0$, it is a demand node with a demand of $-\mathbf{s}_i$. If $-\mathbf{s}_i = 0$, we categorize node $\mathbf{v}_i$ as a transshipment node. Besides, considering each edge $e_{ij} \in \mathcal{E}$, $\mathbf{D}_{ij}$ is the distance of the edge $e_{ij}$ and we set $\mathbf{D}_{ii} \to +\infty$ to prevent self-transportation of nodes here. In comparison to traditional vanilla OT, our OFT considers flow balance constraints $\mathbf{P}\mathbf{1}_N - \mathbf{P}^\top \mathbf{1}_N = \mathbf{s}$ instead of the marginal constraints in $U(\mathbf{a}, \mathbf{b})$ in Eq. 1.

**Exact Solving for OFT** For solving the optimization in Eq. 2, in the field of OT, solutions are typically obtained indirectly. Specifically, one can first define the cost matrix using the geodesic distance (or shortest path metric):

$$\mathbf{C}_{ij} = \min_{K \geq 0, (i_k)_k : i \to j} \Big\{ \sum_{k=1}^{K-1} D_{i_k, i_{k+1}}, 0 \leq k \leq K, e_{i_k, i_{k+1} \in \mathcal{E}} \Big\} \tag{3}$$

where $i \to j$ indicates $i_1 = i$ and $i_K = j$, which is the path starts at $i$ and ends at $j$. The primary formulation of OT, or called the 1-Wasserstein distance can be formulated as

$$\min_{\pi \geq \mathbf{0}} < \mathbf{C}, \pi > \quad \text{s.t.} \quad \pi \mathbf{1}_N = \mathbf{a}, \pi^\top \mathbf{1}_N = \mathbf{b} \tag{4}$$

where **pi** represents the final transportation results between the source and target nodes, which can be equivalent to $\mathbf{P}$ given the routing of the shortest path in Eq. 3. Then one can adopt the Network Simplex (Dantzig, 1951) or Sinkhorn Algorithms (Cuturi, 2013) to get the solutions. However, these shortest path-based approachs have some notable limitations because the routes in the graph are fixed, and there is no way to impose constraints on the flow of any arbitrary edge or node, which limits the potential for real applications such as real traffic scenarios. Another line of approaches (Li et al., 2010; Mohammad Ebrahim & Razmi, 2009) is to use heuristic algorithms based on minimum cost flow. However, different from the algorithms of transportation problems, most of these algorithms assume that all data are integers (e.g. (de Vos, 2023)) and aim to find an integer-valued flow with the minimum total cost while meeting the supply-demand constraints for the graph. This severely limits the algorithms' applications on high-precision data.

### 3.2 Optimal Flow Transport with Entropic Regularization

In this subsection, we propose the definition of entropic OFT and corresponding OFT-Sinkhorn to get the approximate solution for OFT.

**Formulation for Entropic OFT** Differing from previous CPU-based algorithms, in this paper, following (Cuturi, 2013), we consider the entropic OFT and employ a GPU-friendly matrix-vector iterative algorithm to speed up the computations for OFT. However, directly adding entropy regularization cannot derive an algorithm to obtain an approximate solution as directly as vanilla OT. The reason is that: 1) there are many isolated points that do not have any flow passing through them in the exact solution, while under entropy regularization, flows are necessarily channeled through them, causing significant bias; 2) the flow balance constraint cannot directly form an alternating iterative algorithm like

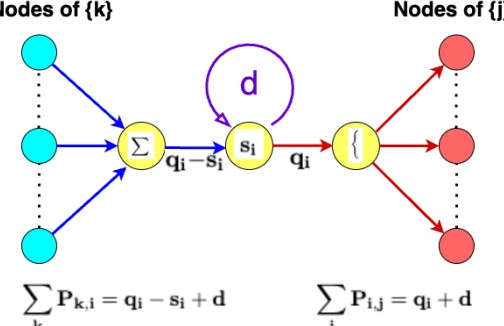

Figure 2: **Illustration of our constraints in Entropic OFT.** The flow constraints satisfied by Eq. 5, where each node includes virtual flow($\mathbf{d}$), input flow($\mathbf{q_i}$), and output flow($\mathbf{q_i} - \mathbf{s_i} + \mathbf{d}$).

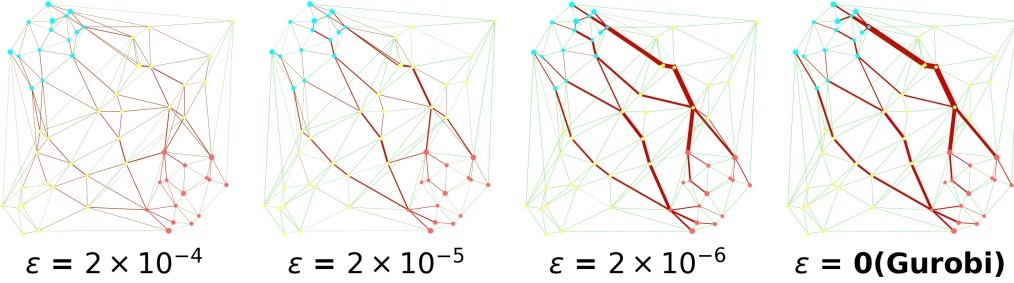

$$\epsilon = 2 \times 10^{-4} \qquad \epsilon = 2 \times 10^{-5} \qquad \epsilon = 2 \times 10^{-6} \qquad \epsilon = 0(\text{Gurobi})$$

Figure 3: Visualization of network flow results by varying the regularization coefficient $\epsilon$ for OFT. We observe that as $\epsilon$ decreases, our approximate solution calculated by EOFT-Sinkhorn approaches closer to the exact solution computed by Gurobi.

the marginal constraint. To address the issues mentioned above, we have modified the optimization objectives and constraints:

$$\min_{\mathbf{P},\mathbf{q}} < \mathbf{D}, \mathbf{P} > -\epsilon H(\mathbf{P}) \quad \text{s.t.} \quad \mathbf{P}\mathbf{1}_N = \mathbf{q} + \mathbf{d}, \mathbf{P}^\top \mathbf{1}_N = \mathbf{q} - \mathbf{s} + \mathbf{d}, \text{Diag}(\mathbf{P}) = \mathbf{d} \tag{5}$$

where $\mathbf{d} = d \cdot \mathbf{1}_N$ represents the virtual flow from each node to itself, given the constant $d$. Here, $\mathbf{D}$ is the distance matrix and we set $\mathbf{D}_{ii} = 0$ for every node $i$ and increase the constraint $\mathbf{P}_{ii} = d$, ensuring that each node, including the isolated nodes mentioned earlier, carries at least $d$ units of flow, which reduces the impact of isolated points for the solutions. Additionally, $\mathbf{q}$ represents the flow out of the nodes (excluding the virtual flow). Then given the net outflow $\mathbf{s}$, we can infer the flow into each node as $\mathbf{q} - \mathbf{s}$. The purpose of these transformations above is to establish a form similar to the marginal constraints in line with vanilla OT, thereby enabling the derivation of a Sinkhorn-like algorithm to obtain an approximate solution for OFT. Based on the method of Lagrange multipliers, we can derive the following proposition (see the proof in Appendix C.1), which aids us in further understanding the prosperity of the solution for OFT.

**Proposition 2.** *The solution to Eq. 5 is unique and has the form:*

$$\mathbf{P} = \text{Diag}(\mathbf{u})\mathbf{K}\text{Diag}(\mathbf{v}) \quad \text{and} \quad \mathbf{q} = \frac{\mathbf{s}}{2} + \left( (\mathbf{K}\mathbf{v}) \cdot (\mathbf{K}^\top \mathbf{u}) + \frac{\mathbf{s}^2}{4} \right)^{\frac{1}{2}} - \mathbf{d} \tag{6}$$

*Here $\mathbf{u}, \mathbf{v}$ are two (unknown) scaling variables satisfying $\mathbf{u} \odot \mathbf{v} = \mathbf{1}$ and $\mathbf{K} = e^{(-\mathbf{C}+diag(\mathbf{h}))/\epsilon}$ where $\mathbf{h}$ is defined as:*

$$\mathbf{h} = \epsilon(\log \mathbf{d} - \log \mathbf{u} - \log \mathbf{v}) \tag{7}$$

Similar to the entropic OT, the solution $\mathbf{P}$ still satisfies $\mathbf{P} = \text{diag}(\mathbf{u})\mathbf{K}\text{diag}(\mathbf{v})$. However, the output flow $\mathbf{q}$ needs to be calculated, and the Gibbs kernel $\mathbf{K}$ is no longer a fixed matrix. Therefore, the iteration process appears more complex compared to the vanilla Sinkhorn algorithm, which we will discuss in detail next.

**OFT-Sinkhorn Algorithm** Here we propose the OFT-Sinkhorn algorithm 3.2, in which we aim to get the optimal solution of Eq. 5 through matrix-vector iterations. An intuitive idea is to iteratively update $\mathbf{P}$, $\mathbf{q}$, and $\mathbf{K}$ to obtain the optimal solution. For the coupling $\mathbf{P}$, based on the solution form $\mathbf{P} = \text{Diag}(\mathbf{u})\mathbf{K}\text{Diag}(\mathbf{v})$ and the marginal constraints in Eq. 5, we derive the following iterations for $\mathbf{u}^{(l)}$ and $\mathbf{v}^{(l)}$ with the $(l+1)-$th iteration:

$$\mathbf{u}^{(l+1)} = \frac{\mathbf{q}^{(l)} + \mathbf{d}}{\mathbf{K}^{(l)}\mathbf{v}^{(l)}}, \quad \mathbf{v}^{(l+1)} = \frac{\mathbf{q}^{(l)} - \mathbf{s} + \mathbf{d}}{(\mathbf{K}^{(l)})^\top \mathbf{u}^{(l+1)}} \tag{8}$$

---

**Algorithm 1** OFT-Sinkhorn: Sinkhorn-based Algorithm for Entropic Optimal Flow Transport

**Input:** Distance Matrix $\mathbf{D}$, Marginal Difference $\mathbf{s}$, maximum iteration number $L$, Virtual-flow $\mathbf{d}$.

**Output:** The Flow Matrix $\mathbf{P}^*$

    Initialize $\mathbf{K} = \exp\left(-\frac{\mathbf{D}}{\epsilon}\right)$ element-wise.

    **for** $l = 0$ **to** $L - 1$ **do**

        Update $u^{(l+1)}$ and $v^{(l+1)}$ according to Eq. 8

        Update $\mathbf{K}^{(l+1)}$ by Eq. 10

        Update $\mathbf{q}^{(l+1)}$ according to Eq. 9

    **end for**

    Compute $\mathbf{P} = \text{Diag}(\mathbf{u}) \cdot \mathbf{K} \cdot \text{Diag}(\mathbf{v})$

    **return** $\mathbf{P}^* = \mathbf{P} - \mathbf{P}^\top$

---

where we initialize $\mathbf{v}^{(0)} = \mathbf{1}$, $\mathbf{K}^{(0)} = e^{-\mathbf{D}/\epsilon}$, and $\mathbf{q}^{(0)} = \max\{\tau, \tau + \mathbf{s}\}$ where $\tau$ is a sufficiently small positive value. By initializing in this manner, we can ensure that the two marginals of the

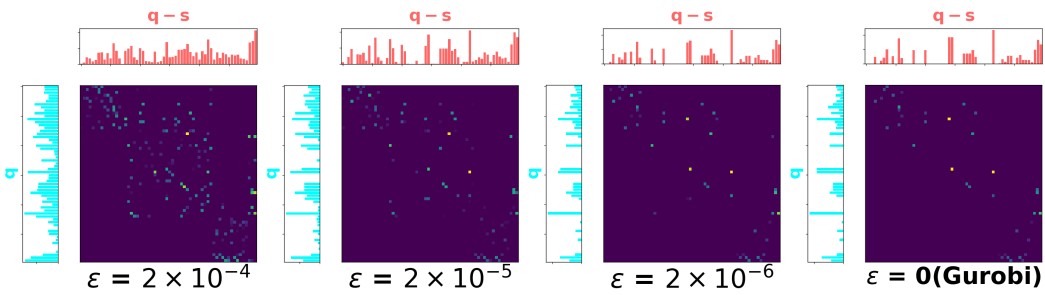

Figure 4: Visualization of the coupling and marginals by varying the regularization coefficient $\epsilon$ for EOFT. As $\epsilon$ decreases, the solutions of the coupling and marginals calculated by EOFT-Sinkhorn become increasingly sparse, approaching the exact solution.

coupling are positive ($\mathbf{q}^{(0)} > \tau$ and $\mathbf{q}^{(0)} - \mathbf{s} > \tau$) , making both $\mathbf{u}$ and $\mathbf{v}$ positive, thereby guaranteeing that $\mathbf{P}$ is positive. Regarding the calculation of $\mathbf{q}$, following Eq. 6, we can update it as:

$$\mathbf{q}^{(l+1)} = \frac{\mathbf{s}}{2} + \left( (\mathbf{K}^{(l+1)}\mathbf{v}^{(l+1)}) \cdot ((\mathbf{K}^\top)^{(l+1)}\mathbf{u}^{(l+1)}) + \frac{\mathbf{s}^2}{4} \right)^{\frac{1}{2}} - \mathbf{d} \tag{9}$$

In fact, for the iteration of $\mathbf{q}^{(l+1)}$, we would need to apply a truncation operation as done for $\mathbf{q}^{(0)}$ to ensure $\mathbf{q}^{(l+1)} \geq \tau$ and $\mathbf{q}^{(l+1)} - \mathbf{s} \geq \tau$. However, in practical iterations, we find that the computed result for $\mathbf{q}^{(l+1)}$ typically naturally satisfies these conditions. Furthermore, for the iteration of $\mathbf{K}$, we can utilize the constraints $\text{Diag}(\mathbf{K}) = \mathbf{d}$ in Eq. 5 to obtain:

$$\text{Diag}(\mathbf{K}^{(l+1)}) = \frac{\mathbf{d}}{\mathbf{u}^{(l+1)} \odot \mathbf{v}^{(l+1)}} \tag{10}$$

and the remaining matrix elements still maintain $\mathbf{K}_{ij}^{(l+1)} = e^{-\mathbf{D}_{ij}/\epsilon}$ for $i \neq j$. By iterating $l = 1, 2, \ldots$ until convergence, we can obtain the optimal solution $\mathbf{P}^*$ for entropic OFT. Similar to how the Sinkhorn algorithm for entropic OT often serves as an activation layer to ensure that the output of neural networks is a doubly stochastic matrix (Wang et al., 2019), our OFT-Sinkhorn algorithm can also serve as a layer to ensure that the output matrix satisfies flow balance constraints, which are typically enforced using loss functions in previous works (Bengio et al., 2023). Furthermore, since the matrix $\mathbf{P}^*$ exhibits backflow $\left(\text{i.e., } \mathbf{P}_{ij}^* > 0 \text{ and } \mathbf{P}_{ji}^* > 0 \text{ for any node indices } i \text{ and } j\right)$, we can perform a backflow removal operation $\mathbf{P}^* \leftarrow \max\{\mathbf{0}, \mathbf{P}^* - (\mathbf{P}^*)^\top\}$, which results in a $\mathbf{P}^*$ closer to the exact solution of OFT.

**Global Convergence** Then we give the convergence discussion. Following (Franklin & Lorenz, 1989), we adopt the Hilbert projective metric to prove our global convergence which is defined as:

$$d_{\mathcal{H}}\left(\mathbf{u}, \mathbf{u}'\right) \stackrel{\text{def.}}{=} \log \max_{i,j} \frac{\mathbf{u}_i \mathbf{u}_j'}{\mathbf{u}_j \mathbf{u}_i'}$$

Then we can get the theoretical results that our OFT-Sinkhorn algorithm has linear convergence and the proof are given in Appendix E.

**Theorem 1.** *The iterative scheme for OFT-Sinkhron algorithm has **linear** convergence. More precisely: one has $\left(\mathbf{u}^{(l)}, \mathbf{v}^{(l)}, \mathbf{K}^{(l)}, \mathbf{q}^{(l)}\right) \to (\mathbf{u}^*, \mathbf{v}^*, \mathbf{K}^*, \mathbf{q}^*)$ and*

$$d_{\mathcal{H}}\left(\mathbf{u}^{(\ell)}, \mathbf{u}^\star\right) = O(\lambda(\mathbf{K})^{2l}) \quad and \quad d_{\mathcal{H}}\left(\mathbf{v}^{(\ell)}, \mathbf{v}^\star\right) = O(\lambda(\mathbf{K})^{2l}) \tag{11}$$

*where*

$$\lambda(\mathbf{K}) = \max_l \lambda(\mathbf{K}^{(l)}) = \sup \left\{ \frac{d_{\mathcal{H}}\left(\mathbf{K}^{(l)}\mathbf{v}, \mathbf{K}^{(l)})\mathbf{v}'\right)}{d_{\mathcal{H}}\left(\mathbf{v}, \mathbf{v}'\right)} \right\} \leq 1 \tag{12}$$

Additionally, we further discuss the numerical convergence of our OFT-Sinkhorn algorithm by varying the iterations in the experimental section.

## 3.3 CAPACITATED OPTIMAL FLOW TRANSPORT WITH ENTROPIC REGULARIZATION

To align with the minimum cost flow problems, in this subsection, we consider adding constraints to our OFT so that our algorithm can address the minimum cost flow problem with matrix-vector iterations. The detailed algorithm is shown in Alg. 2 in Appendix B. Next, we will categorize the capacity constraints into node constraints and edge constraints, and subsequently refine our OFT-Sinkhorn algorithm accordingly.

**Capacitated Constraints on Nodes** Initially, we consider Capacitated constraints on nodes, which involve imposing constraints on the optimization in Eq. 5: $\mathbf{q} \leq \mathbf{r}$ and $\mathbf{q} - \mathbf{s} \leq \mathbf{r}$, where $\mathbf{r}$ represents the maximum capacity for incoming or outgoing flows at nodes. To address these capacity constraints in the OFT-Sinkhorn Algorithm modification, we perform a truncation operation, specifically setting $\mathbf{q}^{l+1} \leftarrow \min\left(\mathbf{r}, \min(\mathbf{r} - \mathbf{s}, \mathbf{q}^{l+1})\right)$, where $\mathbf{q}^{l+1}$ is defined in Eq. 9 for the $(l+1)$-th iteration within the OFT-Sinkhorn Algorithm.

**Capacitated Constraints on Edges** Then we examine the capacitated constraints on edges, which replace the constraints $\mathrm{diag}(\mathbf{P}) = \mathbf{d}$ in Eq. 5 with the constraints $\mathbf{P} \leq \mathbf{S}$, where $\mathbf{S}_{ii} = d$ represents the constraints of virtual flow and $\mathbf{S}_{ij}$ denotes the capacity for edge $e_{ij}$. By employing the Lagrangian method, we can derive the following proposition, with the proof provided in the Appendix C.2.

**Proposition 3.** *The solution of entropic OFT with edge capacity is unique and has the form:*

$$\mathbf{P}^* = \mathrm{Diag}(\mathbf{u})\mathbf{K}\mathrm{Diag}(\mathbf{v}) \quad and \quad \mathbf{q}^* = \frac{\mathbf{s}}{2} + \left((\mathbf{K}\mathbf{v}) \odot (\mathbf{K}^\top \mathbf{u}) + \frac{\mathbf{s}^2}{4}\right)^{\frac{1}{2}} - \mathbf{d} \tag{13}$$

*Here $\mathbf{u}, \mathbf{v}$ are two (unknown) scaling variables satisfying $\mathbf{u} \odot \mathbf{v} = \mathbf{1}$ and $\mathbf{K} = e^{(-\mathbf{D}+\mathbf{G})/\epsilon}$ where $\mathbf{G}$ is defined as*

$$\mathbf{G} = \min\left(\mathbf{D} + \epsilon \cdot \log\left(\mathrm{Diag}(\mathbf{u}^{-1})\mathbf{S}\mathrm{Diag}(\mathbf{v}^{-1})\right), \mathbf{0}\right) \tag{14}$$

For the Capacitated Edges Constraints in the OFT-Sinkhorn algorithm, we will modify the iteration for $\mathbf{K}$ in Eq. 10 as follows:

$$\mathbf{K}^{(l+1)} = \min\left(\mathbf{K}, \mathrm{Diag}(\mathbf{u}^{(l+1)})^{-1}\mathbf{S}\mathrm{diag}(\mathbf{v}^{(l+1)})^{-1}\right) \tag{15}$$

By iteratively applying Eq. 8, Eq. 9, and Eq. 15 until convergence, we can derive the modified OFT-Sinkhorn algorithm based on edge capacity constraints.

Capacitated constraints on nodes and edges are typically meaningful in practical scenarios. For instance, in transportation problems where edges represent roads and nodes denote intersections at either end of the roads, node capacity constraints can effectively express the limit of vehicles passing through intersections. Meanwhile, edge capacity constraints can depict the maximum flow allowed on each road, thereby enabling our capacitated OFT to more accurately model transportation issues within a traffic road network.

## 4 EXPERIMENT

### 4.1 DATASETS AND EXPERIMENT SETTINGS

We conduct experiments using an NVIDIA GeForce RTX 4090 GPU, with programs implemented in Python and PyTorch. We evaluate our approach on three graph datasets: Uniform-MCF , NETGEN-MCF and Vision-MCF. For all experiments except the ablation study, we use the same settings to demonstrate our method's robustness. Since only our matrix iteration algorithm can run efficiently on GPU, the other three comparison methods were tested on the Intel i9-10920X CPU.

**Datasets** We adopt the following datasets in our experiments:

- **Uniform-MCF (Double-precision)**: We consider the symmetry between source and target nodes(i.e., 10% of nodes designated as sources and targets), constructing four synthetic datasets accordingly(w/o constraint, w/edge constraint, w/node constraint, w/edge+node constraint). We utilizes Gaussian function to generate double-precision point within Euclidean space. The initial flow of source and target nodes are generated by Uniform function which is also double-precision. The edge capacity are set to the same value on the same set of parameters. For more details, please refer to Appendix D.

Table 1: Evaluation Results on Uniform-MCF-500, Uniform-MCF-5k and Uniform-MCF-10k. We compared our matrix-vector iterative algorithm (EODT-Sinkhorn) with classical MCF algorithms (such as Real and ZKW) and commonly used general solver Gurobi. One can find that our algorithm is significantly more efficient than traditional algorithms when computing large-scale Maximum Flow instances, all while maintaining low computational errors. The fastest solution time is bolded.

| Size | Methods | w/o constraint | | w/ edge constraint | | w/ node constraint | | w/ edge+node constraints | |
|---|---|---|---|---|---|---|---|---|---|
| | | obj | time | obj | time | obj | time | obj | time |
| $500 \times 500$ | Real | 10.02 | 14.8 s | 10.067 | 101 s | 10.065 | 101 s | 10.066 | 101.9s |
| | ZKW | 10.02 | 18.4 s | 10.067 | 129 s | 10.065 | 139.4 s | 10.066 | 120.8 s |
| | Gurobi | 10.02 | 768 s | 10.067 | 4254 s | 10.066 | 4314 s | 10.067 | 4052 s |
| | ours | 10.11 | **4.18 s** | 10.137 | **34 s** | 10.67 | **43.4 s** | 10.138 | **72.8 s** |
| $5k \times 5k$ | Real | 10.059 | 1843 s | 10.071 | 1036 s | 10.071 | 899 s | 10.064 | 868 s |
| | ZKW | 10.059 | 2034 s | 10.071 | 977s | 10.071 | 1200 s | 10.064 | 1015 s |
| | Gurobi | - | $\geq$ 3 hours | - | $\geq$ 3 hours | - | $\geq$ 3 hours | - | $\geq$ 3 hours |
| | ours | 10.311 | **33 s** | 10.18 | **814 s** | 10.431 | **316 s** | 10.186 | **656 s** |
| $10k \times 10k$ | Real | 10.08 | 3363 s | 10.0898 | 2032 s | 10.0898 | 1842 s | 10.0898 | **1425 s** |
| | ZKW | 10.08 | 3781 s | 10.0898 | 2396 s | 10.0898 | 2543 s | 10.0898 | 1961 s |
| | Gurobi | - | $\geq$ 3 hours | - | $\geq$ 3 hours | - | $\geq$ 3 hours | - | $\geq$ 3 hour |
| | ours | 10.728 | **47 s** | 10.1496 | **1352 s** | 10.18 | **1021 s** | 10.147 | 1462 s |

- **NETGEN (Integer-precision)**: Note that NETGEN (Klingman et al., 1974) is a well-established generator that produces integer-precision instances for the Minimum Cost Flow (MCF) problem and other network optimization problems. It is designed to generate minimum-cost flow problems based on a set of parameters including node size, arc costs, and arc capacities. In our data generation process, we utilize PyNETGEN , a Python implementation of NETGEN. For further details, please refer to Appendix D.

- **Vision (Real Scene)**: This family(Goldberg, 2008) consists of MCF instances based on large-scale maximum flow problems arising in computer vision applications(with over one million nodes). These data files were made available by the Computer Vision Research Group at the University of Western Ontario. The detail of the data is given in Appendix D.

**Baselines.** To demonstrate the feasibility and effectiveness of our OFT-Sinkhorn method, we compare it with representative MCF algorithms (including classic and SOTA algorithms). We consider the following five methods as our baselines: **1)Real( (Papadimitriou & Steiglitz))** is an exact algorithm that combines SPFA (Ahuja et al., 1995) and augmenting paths which is particularly suitable for solving minimum cost flow problems in graphs of fixed distribution of capacity; **2)ZKW( (Goldberg et al.))** is a variant of Successive Shortest Path Algorithm based on augmenting paths and DFS to efficiently find the shortest path. It is particularly suitable for solving minimum cost flow problems in graphs of random distribution of capacity. **3)Gurobi( (Gurobi Optimization, LLC, 2021))** is a powerful optimization solver that can efficiently handle a wide range of mathematical optimization problems, including the minimum cost flow problem. However, as the number of nodes and instances increases, Gurobi's computational efficiency decreases. **4)pns( (Kara & Özturan, 2022))** is one of the SOTA Parallel network simplex algorithm for the MCF problem. **5)lemon( (Király & Kovács, 2012))** is a is a Highly optimized CPU based solver for MCF problems. Here we test on lemon-ns which is the network simplex version of lemon.

## 4.2 Experimental Results

In this section, we discuss the overall performance of various methods on uniform , NETGEN and Vision datasets which is presented in Table 1 , Table 2 and Table 3.

**Evaluation on Uniform Data.** Uniform-MCF is a double-precision MCF instance with predefined edge capacities. In terms of Obj across each dataset, both Real and ZKW consistently achieve the optimal solution, while Gurobi gave a tight approximate solution with a gap of less than 0.1% as shown in Table 1. This might be attributed to the fact that the MCF problem is solvable in polynomial time, where exact algorithms can always provide the optimal solution. However, their efficiency significantly decreases with larger datasets. Notably, Gurobi struggles with problems exceedomg

Table 2: Evaluation of MCF on Small to medium-sized NETGEN dataset.

| Methods | $100 \times 100$ | | $500 \times 500$ | | $1k \times 1k$ | | $5k \times 5k$ | |
|---|---|---|---|---|---|---|---|---|
| | obj | time | obj | time | obj | time | obj | time |
| Real( Papadimitriou & Steiglitz) | 64.58 | 421 ms | 20.418 | 1028 s | 23.365 | 2340 s | 15.652 | 18592 s |
| ZKW( Goldberg et al.) | 64.58 | 504 ms | 20.418 | 706 s | 23.365 | 1639 s | 15.652 | 7491 s |
| Gurobi( Gurobi Optimization, LLC (2021)) | 64.58 | 77.3 s | 20.42 | 4572 s | – | ≥ 3 hours | - | ≥ 10 hours |
| EOFT-Sinkhorn | 65.05 | 22.2 s | 20.548 | 82 s | 24.696 | 420 s | 16.41 | 1261 s |

Table 3: Evaluation of MCF on very large sparse graph.

| Methods | Netgen_8 | | Netgen_lo_8 | | Vision | |
|---|---|---|---|---|---|---|
| | obj | time | obj | time | obj | time |
| pns(k=1,p=1) (Kara & Özturan, 2022) | 18.33 | 335 s | 12.88 | 89 s | 30.16 | **1774 s** |
| pns(k=4,p=4) | 18.33 | 261 s | 12.88 | 71 s | 30.16 | 2579 s |
| pns(k=16,p=16) | 18.33 | 286 s | 12.88 | 91 s | 30.16 | 2512 s |
| lenmon-ns (Királly & Kovács, 2012) | 18.33 | 186 s | 12.88 | **51 s** | 30.16 | 2805 s |
| EOFT-Sinkhorn | 18.83 | **50.4 s** | 13.08 | 59.3 s | 32.04 | 1802 s |

5,000 nodes, which we indicate with a "-" symbol. On w/o constraints, there are only flow balance constraints mentioned in Eq. 5. For problems that incorporate both capacity and node constraints, the objective (obj) of the problem slightly increases (e.g., from 10.02 to 10.06) due to the additional constraints, and the computational time increases almost linearly with the problem size. In contrast, our entropy regularization method, specifically the Sinkhorn algorithm (Cuturi, 2013), offers faster solving speeds due to its GPU-friendliness. The results demonstrate that our algorithm significantly outperforms traditional algorithms in terms of efficiency when computing large-scale MCF instances, while maintaining low computational error gap. For further details and parameters regarding the Uniform-MCF , please refer to Table 4 in Appendix D.

**Evaluation Results on small to medium-size NETGEN.** The experimental details of NETGEN datasets are given in Table 5 and the results are given in Table 2. For smaller problems, Real ((Papadimitriou & Steiglitz)) and ZKW ((Goldberg et al.)) achieve the best solution quality and computational time. However, as the dataset size $|D|$, the total number of nodes $|V|$ and the total number of edges $|E|$ increase, our method demonstrates superior solution speed over former 3 methods. Lemon achieve best solution and speed under all-cases, demonstrated outstanding performance in solving small to medium-scale MCF problems as a SOTA method." Appendix D shows the parameters and details about NETGEN experiments.

**Evaluation Results on very large sparse graph.** The experimental details for large sparse datasets are presented in Table 6, while the corresponding results are summarized in Table 3. To efficiently solve these three exceptionally large sparse MCF problems, we tailored our algorithm to better leverage the triplet structure inherent in sparse graphs. As the pioneering work applying entropic regularization to the MCF problem and developing a GPU-friendly algorithm, our approach effectively tackles the minimum cost flow problem on extremely large sparse graphs. In certain cases, it even surpasses state-of-the-art algorithms in obtaining high-quality approximate solutions.

## 4.3 ABLATION STUDY

Then we will analyze the convergence, effect of regularization, and numerical stability in depth.

**Convergence of EOFT** In Figure 5, we illustrate how the marginal distribution evolves with the number of iterations for an instance with 5000 nodes. The marginal difference is computed as $\mathbf{P1}_N - \mathbf{P}^\top \mathbf{1}_N$ (denoted as $\mathbf{s_{soft}}$), while the Ground Truth (GT) marginals are represented by $\mathbf{s}$. Initially, we generate the flow for the source and target separately using a one-dimensional Gaussian function, denoted as $\mathbf{s_{source}}$ and $\mathbf{s_{target}}$. To estimate the distribution of the source and target marginals, we employ Gaussian Kernel Density Estimation (KDE) (Parzen, 1962), defined as $K(u) = \frac{1}{\sqrt{2\pi}} e^{-\frac{1}{2}u^2}$. The marginal difference at transshipment nodes is obtained by computing the difference between the estimated distributions of the source and target marginals, i.e. $\mathbf{s_{source}} - \mathbf{s_{target}}$. The results demonstrate that as the number of iterations increases, EOFT progressively converges to the GT marginals, validating the accuracy and stability of our algorithm. (e.g., the red dotted line (EOFT-200 iterations) in Figure 5 almost overlaps with the solid black line which is GT marginals).

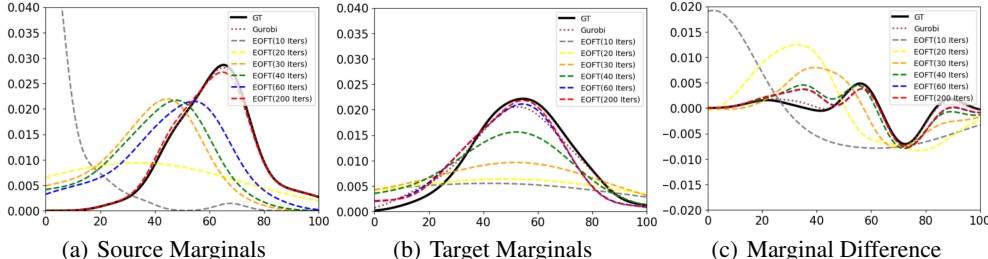

| (a) Source Marginals | (b) Target Marginals | (c) Marginal Difference |

Figure 5: Visualization of the marginals by varying iterations for EOFT. As iterations increase, the marginals calculated by the OFT-Sinkhorn algorithm (dotted line) gradually approach Ground Truth (black solid line).

**Sparsity of the Coupling** In domains such as logistics planning, an exact and sparse transport plan is desired. In this section, we conduct tests on the sparsity of the transport plan for EOFT-Sinkhorn methods with different regularization coefficients $\epsilon$. We uniformly sample a network of 60 nodes, designating 15 as sources and 15 as targets. Figure 3 visualizes the transportation plan generated by the EOFT method. The dotted green lines represent potential node connections, whereas the red solid lines signify the actual transportation routes. The color intensity along these paths is indicative of the flow volume, offering a clear visual illustration of the plan's density. The results reveal that as the of $\epsilon$ decrease, the approximate solution from the EOFT-Sinkhorn method converges more closely to the optimal solution achieved using Gurobi. Similarly, Figure 4 illustrates the transformation of the coupling matrix and marginals as the regularization coefficient $\epsilon$ decreased. With a decreasing $\epsilon$, the optimal coupling matrix exhibits increased sparsity, and the transportation plan $\mathbf{P}$ progressively aligns with the solution delivered by Gurobi.

**Numerical Stability** Note the stability is critical in algorithm design and Figure 6 evaluates the numerical stability of the OFT-Sinkhorn algorithm. Each data point reflects the mean performance of four random instances, with all other parameters kept constant to ensure fair comparison. The heatmap displays both cost and time, where NaN represents a computational failure at that particular parameter combination. The horizontal axis shows the progressively reducing

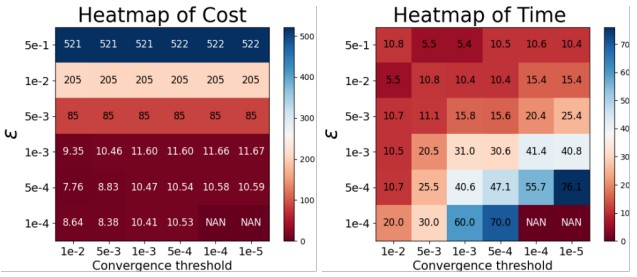

Figure 6: Visualization of the cost and time by varying the $\epsilon$ and Error(convergence threshold) for EOFT. Under most cases, EOFT exhibits a smooth changing process. Each point is obtained by the average of 4 instance

Err (the algorithm's convergence threshold, defined as $\mathbf{P1}_N - \mathbf{P}^\top \mathbf{1}_N - \mathbf{s}$) and the vertical axis illustrates the gradually decreasing $\epsilon$. The findings reveal that as either the convergence threshold or $\epsilon$ diminishes, the EOFT algorithm requires more time to converge, yet it achieves a solution of higher quality. Our algorithm exhibits remarkable stability across a wide range of parameters, notably when $\epsilon$ range from $5 \times 10^{-1}$ to $1 \times 10^{-4}$ and convergence threshold ranges from $1 \times 10^{-2}$ to $1 \times 10^{-5}$.

## 5 CONCLUSION AND FUTURE WORK

This paper has formulated the optimal flow transport problem beyond the vanilla discrete OT, to improve its applicability in real-world problems. We also present new time-efficient methods OFT-Sinkhorn algorithm to solve the problem and its variants, which is solved by GPU-friendly matrix iterations. Experiments show that our method is more efficient than general solvers in solving the minimum cost flow problem, which proves that our EOFT algorithm can be effectively used to solve the minimum cost flow problem. For future work, we will combine our approach with other OT variants, such as partial transportation Phatak et al. (2023), robust optimization Raghvendra et al. (2024), or explore parallel computation of OFT (Lahn et al., 2023).

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

## A  APPENDIX

## B  DETAILS ALGORITHM OF EOFT WITH NODE AND EDGE CONSTRAINTS

---

**Algorithm 2** OFT-Sinkhorn: Iterative Sinkhorn-based Algorithm for (capacitied) Entropic Optimal Flow Transport with Edge and Node Constraints

---

**Input:** Distance Matrix $\mathbf{D}$, Edge Capability $\mathbf{S}$, Node Capability $\mathbf{r}$, Marginal Difference $\mathbf{s}$, maximum iteration number $L$, Virtual-flow $\mathbf{d}$.

**Output:** The Allocation Matrix $\mathbf{P}_{ij}$ for $i, j = 1, \ldots, N$

   Initialize $\mathbf{K} = \exp\left(-\frac{\mathbf{D}}{\epsilon}\right)$ element-wise.

   **for** $l = 0$ **to** $L - 1$ **do**

      Update $\mathbf{u}^{(l+1)}$, $\mathbf{v}^{(l+1)}$ according to Eq. 8

      Update $\mathbf{q}^{(l+1)}$ according to Eq. 13

      Update $\mathbf{G}^{(l+1)}$ according to Eq. 14

      Update $\mathbf{K}^{(l+1)}$ by $\exp\left(\frac{-\mathbf{D} + \mathbf{G}^{(l+1)}}{\epsilon}\right)$

   **end for**

   Compute $\mathbf{P} = \text{Diag}(\mathbf{u}) \cdot \mathbf{K} \cdot \text{Diag}(\mathbf{v})$

   **return** $\mathbf{P}^* = \mathbf{P} - \mathbf{P}^\top$

---

## C  PROOF FOR ALGORITHEM EOFT-SH, EOFT-SH-C

Here we provide the Mathematical proof for Proposition 1 and 2 in main text.

### C.1  WITHOUT CONSTRAINS

$$\min_{\substack{\mathbf{P} \geq \mathbf{0}, \\ \mathbf{q} \geq \mathbf{0}}} \quad <\mathbf{D}, \mathbf{P}> -\epsilon H(\mathbf{P})$$

$$\text{subject to} \quad \mathbf{P}\mathbf{1}_N = \mathbf{q} + \mathbf{d}, \mathbf{P}^\top \mathbf{1}_N = \mathbf{q} - \mathbf{s} + \mathbf{d}$$

$$\text{Diag}(\mathbf{P}) = \mathbf{d} \tag{16}$$

Adding dual variables $\mathbf{f}, \mathbf{g}, \mathbf{h}$, the lagrange equation is then:

$$\mathcal{L} = <\mathbf{D}, \mathbf{P}> -\epsilon H(\mathbf{P}) - <\mathbf{f}, \mathbf{P}\mathbf{1}_N - \mathbf{q} - \mathbf{d}> - <\mathbf{g}, \mathbf{P}^\top \mathbf{1}_N - \mathbf{q} + \mathbf{s} - \mathbf{d}> - <\mathbf{h}, \text{Diag}(\mathbf{P}) - \mathbf{d}> \tag{17}$$

We can get partial differential of lagrange equation with respect to P:

$$\frac{\partial \mathcal{L}}{\partial \mathbf{P_{ij}}} = \mathbf{D_{ij}} + \epsilon \log(\mathbf{P_{ij}}) - \mathbf{f_i} - \mathbf{g_j} = 0 (i \neq j) \tag{18}$$

Then we can get :

$$\mathbf{P}_{ij} = e^{\mathbf{f_i}/\epsilon} e^{-\mathbf{D_{ij}}/\epsilon} e^{\mathbf{g_j}/\epsilon}. \tag{19}$$

Similarly:

$$\frac{\partial \mathcal{L}}{\partial \mathbf{P_{ii}}} = \mathbf{D_{ii}} + \epsilon \log(\mathbf{P_{ii}}) - \mathbf{f_i} - \mathbf{g_i} - \mathbf{h_i} = 0 \tag{20}$$

Then we can get :

$$\mathbf{P}_{ii} = e^{\mathbf{f_i}/\epsilon} e^{-(\mathbf{D_{ii}} - \mathbf{h_i})/\epsilon} e^{\mathbf{g_i}/\epsilon}. \tag{21}$$

Thus for the whole matrix $\mathbf{P}$, we can get :

$$\mathbf{P} = \text{Diag}(e^{\mathbf{f}/\epsilon}) e^{-\mathbf{D} + \text{Diag}(\mathbf{h})/\epsilon} \text{Diag}(e^{\mathbf{g}/\epsilon}). \tag{22}$$

Letting $\mathbf{u} = e^{\mathbf{f}/\varepsilon}$, $\mathbf{v} = e^{\mathbf{g}/\varepsilon}$, $\mathbf{K} = e^{-\mathbf{D} + \mathbf{Diag}(\mathbf{h})/\epsilon}$, we can get the optimal solution from $\mathbf{P} = \text{Diag}(\mathbf{u})\mathbf{K}\text{Diag}(\mathbf{v})$.

We can get partial differential of lagrange equation with respect to q:

$$\frac{\partial \mathcal{L}}{\partial \mathbf{q}} = \mathbf{f} + \mathbf{g} = \mathbf{0}. \tag{23}$$

Thus we can get $\mathbf{f} = -\mathbf{g}$    i.e: $\mathbf{u} \odot \mathbf{v} = \mathbf{1}$

Knowing $\mathbf{P1}_N = \mathbf{q} + \mathbf{d}$ and $\mathbf{P}^\top \mathbf{1}_N = \mathbf{q} - \mathbf{s} + \mathbf{d}$, we can get:

$$\textcircled{1} \quad \mathbf{u} = \frac{\mathbf{q} + \mathbf{d}}{\mathbf{Kv}}, \quad \mathbf{v} = \frac{\mathbf{q} - \mathbf{s} + \mathbf{d}}{\mathbf{K}^\top \mathbf{u}} \tag{24}$$

Thus we have :

$$\frac{\mathbf{q} + \mathbf{d}}{\mathbf{Kv}} \odot \frac{\mathbf{q} - \mathbf{s} + \mathbf{d}}{(\mathbf{K})^\top \mathbf{u}} = \mathbf{1}$$

Then we can get the solution of q as :

$$\textcircled{2} \quad \mathbf{q}^* = \frac{\mathbf{s}}{2} + \left( (\mathbf{Kv}) \odot (\mathbf{K}^\top \mathbf{u}) + \frac{\mathbf{s}^2}{4} \right)^{\frac{1}{2}} - \mathbf{d}. \tag{25}$$

Knowing $\mathrm{Diag}(\mathbf{P}) = \mathbf{d}$, we can get that :

$$\mathrm{Diag}(\mathbf{P}) = \mathbf{u_i} \cdot e^{(-\mathbf{D_{ii}} + \mathbf{Diag}(\mathbf{h_i}))/\epsilon} \cdot \mathbf{v_i} = \mathbf{d}$$

So the solution of $\mathbf{d}$ is :

$$\textcircled{3} \quad \mathbf{h} = \epsilon \times (\log \mathbf{d} - \log \mathbf{u} - \log \mathbf{v}) + \mathrm{Diag}(\mathbf{D})$$

From the above derivation, we can know:

$$\textcircled{4} \quad \mathbf{K} = e^{-\mathbf{D} + \mathbf{Diag}(\mathbf{h})/\epsilon}$$

By iterating through $\textcircled{1}, \textcircled{2}, \textcircled{3}$, and $\textcircled{4}$, we obtain the final Proposition

## C.2 WITH CONSTRAINS

$$\begin{aligned} \min_{\substack{\mathbf{P} \geq \mathbf{0}, \\ \mathbf{q} \geq \mathbf{0}}} \quad & <\mathbf{D}, \mathbf{P}> -\epsilon H(\mathbf{P}) \\ \text{subject to} \quad & \mathbf{P1}_N = \mathbf{q} + \mathbf{d}, \mathbf{P}^\top \mathbf{1}_N = \mathbf{q} - \mathbf{s} + \mathbf{d} \\ & \mathbf{P} \leq \mathbf{S}, \mathbf{q} \leq \mathbf{r} \end{aligned} \tag{26}$$

Adding dual variables $\mathbf{f}, \mathbf{g}, \mathbf{H}(\mathbf{H} \geq \mathbf{0})$, the lagrange equation is then:

$$\mathcal{L} = <\mathbf{D}, \mathbf{P}> -\epsilon H(\mathbf{P}) - <\mathbf{f}, \mathbf{P1}_N - \mathbf{q} - \mathbf{d}> - <\mathbf{g}, \mathbf{P}^\top \mathbf{1}_N - \mathbf{q} + \mathbf{s} - \mathbf{d}> - <\mathbf{H}, \mathbf{P} - \mathbf{S}> \tag{27}$$

The partial differential of lagrange equation with respect to $\mathbf{P}$:

$$\frac{\partial \mathcal{L}}{\partial \mathbf{P_{ij}}} = \mathbf{D_{ij}} + \epsilon \log (\mathbf{P_{ij}}) - \mathbf{f_i} \mathbf{1}^\top - \mathbf{1}^\top \mathbf{g_j} - \mathbf{H_{ij}} = 0 \tag{28}$$

Thus we get :

$$\mathbf{P}_{ij} = e^{\mathbf{f_i}/\epsilon} e^{(-\mathbf{D_{ij}} + \mathbf{H_{ij}})/\epsilon} e^{\mathbf{g_j}/\epsilon} = \mathrm{Diag}(\mathbf{u}) \mathbf{K} \mathrm{Diag}(\mathbf{v}) \tag{29}$$

where $\mathbf{u} = e^{\mathbf{f}/\varepsilon}, \mathbf{v} = e^{\mathbf{g}/\varepsilon}, \mathbf{K} = e^{(-\mathbf{D} + \mathbf{H})/\epsilon}$.

We can get partial differential of lagrange equation with respect to q:

$$\frac{\partial \mathcal{L}}{\partial \mathbf{q}} = \mathbf{f} + \mathbf{g} = \mathbf{0}. \tag{30}$$

Thus we can get $\mathbf{f} = -\mathbf{g}$    i.e: $\mathbf{u} \odot \mathbf{v} = \mathbf{1}$ Similar to the Section 3.1:

$$\mathbf{u} \odot \mathbf{v} = \frac{\mathbf{q} + \mathbf{d}}{\mathbf{K}\mathbf{v}} \odot \frac{\mathbf{q} - \mathbf{s} + \mathbf{d}}{(\mathbf{K})^\top \mathbf{u}} = \mathbf{1}$$

Due to the constraint on $\mathbf{q}$:

$$\mathbf{q}^* = \min\left(\frac{\mathbf{s}}{2} + \left((\mathbf{K}\mathbf{v}) \odot (\mathbf{K}^\top \mathbf{u}) + \frac{\mathbf{s}^2}{4}\right)^{\frac{1}{2}} - \mathbf{d}, \mathbf{r}\right) \tag{31}$$

Due to $\mathbf{P} \leq \mathbf{S}$ ,we can get that :

$$\mathbf{P} = \mathrm{Diag}(\mathbf{u})e^{(-\mathbf{D}+\mathbf{H})/\epsilon}\mathrm{Diag}(\mathbf{v}) \leq \mathbf{S} \tag{32}$$

Thus: $\mathbf{H} \leq \mathbf{D} + \epsilon \cdot \log\left(\mathrm{Diag}(\mathbf{u}^{-1})\mathbf{S}\mathrm{Diag}(\mathbf{v}^{-1})\right)$ Thus we can get that :

$$\mathbf{H} = \min\left(\mathbf{D} + \epsilon \cdot \log\left(\mathrm{Diag}(\mathbf{u}^{-1})\mathbf{S}\mathrm{Diag}(\mathbf{v}^{-1})\right), \mathbf{0}\right) \tag{33}$$

## D DETAILS ABOUT EXPERIMENTS

### D.1 DETAILS ABOUT PARAMETERS

Table 4 show the parameters of MCF experiments at Table 1 in main text ,including the Batch size, Regularization coefficients, Split of the node set, $\mathbf{d}$ (Virtual-flow), $\mathbf{Node\_C}$ (Capacity for flows at nodes), $\mathbf{Edge\_C}$ (Capacity for edges)), and $\mathbf{err}$ (Convergence threshold of the iterations).

Table 4: The parameters for experiments on synthetic dataset

| Instance | Instance_Num | $\epsilon$ | Node_Split | d0 | Edge_C | Node_C | Convergence threshold |
|---|---|---|---|---|---|---|---|
| Node_500_w/o constraint | 64 | 5e-4 | 50_400_50 | 1e-3 | - | - | 1e-6 |
| Node_500_w/edge constraint | 256 | 1e-4 | 50_400_50 | 1e-4 | 0.05 | - | 1e-5 |
| Node_500_w/node constraint | 256 | 5e-4 | 50_400_50 | 1e-4 | - | 0.1 | 1e-5 |
| Node_500_w/ edge+node constraints | 256 | 1e-4 | 50_400_50 | 1e-4 | 0.1 | 0.5 | 1e-5 |
| Node_5k_w/o constraint | 16 | 5e-4 | 500_4000_500 | 1e-4 | - | - | 1e-6 |
| Node_5k_w/edge | 16 | 5e-4 | 500_4000_500 | 1e-4 | 0.05 | - | 1e-5 |
| Node_5k_w/node | 8 | 1e-3 | 500_4000_500 | 1e-4 | - | 0.1 | 1e-5 |
| Node_5k_w/ edge+node constraints | 8 | 5e-4 | 500_4000_500 | 1e-4 | 0.1 | 0.5 | 1e-5 |
| Node_10k_w/o constraint | 4 | 5e-4 | 1000_8000_1000 | 1e-6 | - | - | 1e-6 |
| Node_10k_w/ edge constraints | 2 | 5e-4 | 1000_8000_1000 | 1e-4 | 0.05 | - | 1e-5 |
| Node_10k_w/ node constraints | 2 | 5e-4 | 1000_8000_1000 | 1e-4 | - | 0.1 | 1e-5 |
| Node_10k_w/ edge+node constraints | 2 | 5e-4 | 1000_8000_1000 | 1e-4 | 0.1 | 0.5 | 1e-5 |

Similarly, Table 5 show the parameters of MCF experiments at Table 2 in main text, including the Batch size, Regularization coefficients, Split of the node set, $\mathbf{d}$ (Virtual-flow), $\mathbf{Cap\_Range}$ (Capacity range for NETGEN to randomly generate), $\mathbf{Arc\_Num}$ (Totak arc number)), and $\mathbf{err}$ (Convergence threshold of the iterations).

The Convergence threshold $\mathbf{err}$ is defined as the average differance of the marginals:

$$err = \frac{\sum_{i=1}^{\text{batch\_size}} \left\|\mathbf{P}_i \mathbf{1}_n - \mathbf{P}_i^\top \mathbf{1}_m - \mathbf{s}_i\right\|}{\text{batch\_size}}$$

Table 5: The parameters for experiments on NETGEN dataset

| Instance | Instance_Num | $\epsilon$ | Node_Split | d0 | Cap_Range | Arc_Num | err |
|---|---|---|---|---|---|---|---|
| Node_100_NETGEN | 128 | 1e-3 | 10_80_10 | 1e-4 | [0.5,1] | 800 | 1e-5 |
| Node_500_NETGEN | 256 | 5e-4 | 50_400_50 | 1e-4 | [0.5,1] | 64 k | 1e-4 |
| Node_1k_NETGEN | 128 | 5e-4 | 100_800_100 | 1e-4 | [0.5,1] | 80 k | 1e-4 |
| Node_5k_NETGEN | 8 | 5e-4 | 500_4000_500 | 1e-4 | [0.5,1] | 10000 k | 1e-4 |

Table 7 shows the Sparsity degree of the Table 1,including the Cost Matrix, Guroobi and EOFT.The results shows that the Uniform-dataset is sparse graph, and both Gurobi and EOFT have obtained nearly equally sparse solutions across various constraint scenarios and problem sizes. The arc number of NETGEN dataset is about $O(n^2)$ on 500/1k/5k instances, which means they are Dense graph.

Table 6: The parameters for very large sparse graph. For netgen generator, netgen_8 is a sparse graph with a degree 8, and netgen_lo are variants with lower supplies. Vision we adopt the Vision_inv_05 instance mentioned in Kara & Özturan (2022)

| Instance | Instance_Num | Node Num | Arc Num | Total Supply | Capacity_Range | Cost_Range | err | $\epsilon$ | d0 |
|---|---|---|---|---|---|---|---|---|---|
| Netgen_8 | 1 | 1048576 | 8388608 | 1024000 | [1,1000] | [1,10000] | 1e-3 | 1e-2 | 1e-4 |
| Netgen_lo_8 | 1 | 1048576 | 8388608 | 10240 | [1,1000] | [1,10000] | 1e-3 | 1e-2 | 1e-4 |
| Vision | 1 | 3899394 | 23091149 | 10000 | [0,100] | $\frac{[9000,11000]}{capacity}$ | 1e-3 | 1e-2 | 1e-4 |

Table 7: The sparsity degree of the solutions

| Instance | Cost_Mat | Gurobi_Sol | EOFT_Sol |
|---|---|---|---|
| Node_500_w/o constraint | 0.02 | 0.998 | 0.996 |
| Node_500_w/ edge+node constraints | 0.02 | 0.999 | 0.992 |
| Node_5k_w/o constraint | 0.002 | 0.998 | 0.997 |
| Node_5k_w/ edge+node constraints | 0.002 | 0.982 | 0.951 |
| Node_10k_w/o constraint | 0.002 | 0.999 | 0.999 |
| Node_10k_w/ edge+node constraints | 0.002 | 0.962 | 0.934 |

## D.2 Details about Dataset

Uniform-MCF: The average out-degree of this dataset is 16, indicating that Uniform-MCF is a high-precision and sparse graph dataset. The Distance matrix here is generated by gaussian function within $[0, 1]^2$ Euclidean space

NETGEN (Klingman et al., 1974) was used to generate random minimum-cost flow, maximum flow, and assignment problems, exported in DIMACS graph format. We use PyNETGEN as a Python implementation of NETGEN in our data generating process.

PyNETGEN are capable of generating minimum-cost network flows problems according to a set of tuneable parameters that control things like the size of the network and the acceptable ranges of arc costs and capacities. It begins by defining source and sink nodes and randomly distributing supply among them. It then generates a set of "skeleton arcs" to create paths from the sources to the sinks. Skeleton arcs are guaranteed to have enough capacity to carry all required flow, ensuring that the problem instance is feasible. After the skeleton is defined, arcs are randomly generated between pairs of randomly-selected nodes until the desired density is reached The main parameters used for the NETGEN are as follows:

- `nodes` – number of nodes (default 10)

- `sources` – number of source nodes (default 3)

- `sinks` – number of sink nodes (default 3)

- `density` – number of arcs (shown in Table 4)

- `mincost` – minimum arc cost (we set as 10)

- `maxcost` – maximum arc cost (we set as 100)

- `supply` – total supply (we set as 10000)

- `capacitated` – percent of skeleton arcs (0-100) that are capacitated (we set as 100)

- `mincap` – minimum arc capacity (shown in Table 4)

- `maxcap` – maximum arc capacity (shown in Table 4)

## E Convergence of OFT-Sinkhron Algorithm.

Based on the OFT-Sinkhorn algorithm, the corresponding iterative scheme is as follows,

$$\mathbf{u}^{(l+1)} = \frac{\mathbf{q}^{(l)} + \mathbf{d}}{\mathbf{K}^{(l)}\mathbf{v}^{(l)}},$$

$$\mathbf{v}^{(l+1)} = \frac{\mathbf{q}^{(l)} - \mathbf{s} + \mathbf{d}}{(\mathbf{K}^{(l)})^\top \mathbf{u}^{(l+1)}},$$

$$\text{Diag}(\mathbf{K}^{(l+1)}) = \frac{\mathbf{d}}{\mathbf{u}^{(l+1)} \odot \mathbf{v}^{(l+1)}},$$

$$\mathbf{q}^{(l+1)} = \frac{\mathbf{s}}{2} + \left( (\mathbf{K}^{(l+1)}\mathbf{v}^{(l+1)}) \odot ((\mathbf{K}^\top)^{(l+1)}\mathbf{u}^{(l+1)}) + \frac{\mathbf{s}^2}{4} \right)^{\frac{1}{2}} - \mathbf{d},$$

$$(34)$$

for $l = 0, 1, \ldots, M$. The stopping criteria are chose as $\max \left| \mathbf{q}^{(M)} - \mathbf{q}^{(M-1)} \right| < \delta$ with a small error criteria $\delta$. Before analyzing the global convergence of OFT-Sinkhorn algorithm, we introduce the Hilbert projection metric, which is defined as,

$$d_{\mathcal{H}}\left(\mathbf{u}, \mathbf{u}'\right) = \log \max_{i,j} \frac{\mathbf{u}_i \mathbf{u}'_j}{\mathbf{u}_j \mathbf{u}'_i}. \tag{35}$$

**Lemma 1.** *For matrix $\mathbf{K}$, vector $\mathbf{v}$ and $\mathbf{v}'$, the following inequality holds*

$$d_{\mathcal{H}}\left(\mathbf{K}\mathbf{v}, \mathbf{K}\mathbf{v}'\right) \le \lambda(\mathbf{K}) d_{\mathcal{H}}\left(\mathbf{v}, \mathbf{v}'\right) \tag{36}$$

*with*

$$\begin{cases} \lambda(\mathbf{K}) = \frac{\sqrt{\eta(\mathbf{K})}-1}{\sqrt{\eta(\mathbf{K})}+1} < 1, \\ \eta(\mathbf{K}) = \max_{i,j,k,\ell} \frac{\mathbf{K}_{i,k}\mathbf{K}_{j,\ell}}{\mathbf{K}_{j,k}\mathbf{K}_{i,\ell}}. \end{cases} \tag{37}$$

The above theoretical results are given in Theorem 4.1 in the paper (Peyre & Cuturi, 2019) and we use it to prove the following theorem to show our convergence.

**Theorem 2.** *The iterative scheme Eq. 34 for OFT-Sinkhron algorithm has linear convergence. More precisely: one has $\left(\mathbf{u}^{(l)}, \mathbf{v}^{(l)}, \mathbf{K}^{(l)}, \mathbf{q}^{(l)}\right) \to (\mathbf{u}^*, \mathbf{v}^*, \mathbf{K}^*, \mathbf{q}^*)$ and*

$$d_{\mathcal{H}}\left(\mathbf{u}^{(\ell)}, \mathbf{u}^\star\right) = O(\lambda(\mathbf{K})^{2l}),$$

$$d_{\mathcal{H}}\left(\mathbf{v}^{(\ell)}, \mathbf{v}^\star\right) = O(\lambda(\mathbf{K})^{2l}),$$

$$(38)$$

*where*

$$\lambda(\mathbf{K}) = \max_l \lambda(\mathbf{K}^{(l)}) = \sup \left\{ \frac{d_{\mathcal{H}}\left(\mathbf{K}^{(l)}\mathbf{v}, \mathbf{K}^{(l)}\mathbf{v}'\right)}{d_{\mathcal{H}}\left(\mathbf{v}, \mathbf{v}'\right)} \right\} \le 1. \tag{39}$$

*Proof.* For any $\mathbf{v}$ and $\mathbf{v}'$, we have

$$d_{\mathcal{H}}\left(\mathbf{v}, \mathbf{v}'\right) = d_{\mathcal{H}}\left(\mathbf{v}/\mathbf{v}', \mathbf{1}_N\right) = d_{\mathcal{H}}\left(\mathbf{1}_N/\mathbf{v}, \mathbf{1}_N/\mathbf{v}'\right). \tag{40}$$

Then,

$$\begin{aligned} d_{\mathcal{H}}\left(\mathbf{u}^{(l+1)}, \mathbf{u}^*\right) &= d_{\mathcal{H}}\left(\frac{\mathbf{q}^{(l)} + \mathbf{d}}{\mathbf{K}^{(l)}\mathbf{v}^{(l)}}, \frac{\mathbf{q}^* - \mathbf{s} + \mathbf{d}}{\mathbf{K}^*\mathbf{v}^*}\right) \\ &= d_{\mathcal{H}}\left(\frac{\mathbf{K}^{(l)}\mathbf{v}^{(l)}}{\mathbf{q}^{(l)} + \mathbf{d}}, \frac{\mathbf{K}^*\mathbf{v}^*}{\mathbf{q}^* - \mathbf{s} + \mathbf{d}}\right) \\ &\le \max(\lambda(\mathbf{K}^l), \lambda(\mathbf{K}^*)) d_{\mathcal{H}}\left(\frac{\mathbf{v}^{(l)}}{\mathbf{q}^{(l)} + \mathbf{d}}, \frac{\mathbf{v}^*}{\mathbf{q}^* - \mathbf{s} + \mathbf{d}}\right) \\ &\le \lambda(\mathbf{K}) d_{\mathcal{H}}\left(\frac{\mathbf{v}^{(l)}}{\mathbf{q}^{(l)} + \mathbf{d}}, \frac{\mathbf{v}^*}{\mathbf{q}^* - \mathbf{s} + \mathbf{d}}\right), \end{aligned} \tag{41}$$

Lemma 1 is used in Eq. 41 which implies $\mathbf{u}^{(l)} \to \mathbf{u}^*, \frac{\mathbf{v}^{(l)}}{\mathbf{q}^{(l)}} \to \frac{\mathbf{v}^*}{\mathbf{q}^*}$.

Furthermore,

$$
\begin{aligned}
d_{\mathcal{H}}\left(\mathbf{v}^{(l+1)}, \mathbf{v}^*\right) &= d_{\mathcal{H}}\left(\frac{\mathbf{q}^{(l)}+\mathbf{d}}{\mathbf{K}^{(l)}\mathbf{u}^{(l+1)}}, \frac{\mathbf{q}^*-\mathbf{s}+\mathbf{d}}{\mathbf{K}^*\mathbf{u}^*}\right) \\
&= d_{\mathcal{H}}\left(\frac{\mathbf{K}^{(l)}\mathbf{u}^{(l+1)}}{\mathbf{q}^{(l)}+\mathbf{d}}, \frac{\mathbf{K}^*\mathbf{u}^*}{\mathbf{q}^*-\mathbf{s}+\mathbf{d}}\right) \\
&\leq \lambda(\mathbf{K})d_{\mathcal{H}}\left(\frac{\mathbf{u}^{(l+1)}}{\mathbf{q}^{(l)}+\mathbf{d}}, \frac{\mathbf{u}^*}{\mathbf{q}^*-\mathbf{s}+\mathbf{d}}\right)
\end{aligned}
\tag{42}
$$

Lemma 1 is used again in Eq. 42 which shows that $\mathbf{v}^{(l)} \to \mathbf{v}^*, \frac{\mathbf{u}^{(l)}}{\mathbf{q}^{(l)}} \to \frac{\mathbf{u}^*}{\mathbf{q}^*}$.

Substituting $\mathbf{u}^*$ and $\mathbf{v}^*$ into Eq. 34, which deduces to

$$
\text{Diag}(\mathbf{K}^*) = \frac{\mathbf{d}}{\mathbf{u}^* \odot \mathbf{v}^*}, \quad \mathbf{q}^* = \frac{\mathbf{s}}{2} + \left((\mathbf{K}^*\mathbf{v}^*) \odot ((\mathbf{K}^\top)^*\mathbf{u}^*) + \frac{\mathbf{s}^2}{4}\right)^{\frac{1}{2}} - \mathbf{d}.
\tag{43}
$$

$\square$

