# OpenReview forum: "Optimal Flow Transport and its Entropic Regularization: a GPU-friendly Matrix Iterative Algorithm for Flow Balance Satisfaction"
_ICLR.cc/2025/Conference — ICLR 2025 Poster_

### Official Review · Reviewer_T4Wu · 2024-10-17

**Soundness:** 3
**Presentation:** 3
**Contribution:** 3
**Rating:** 6
**Confidence:** 4

**Summary:**

This work extends optimal transport to network flow problems and formulated an entropic algorithm.

**Strengths:**

1. The writing is clear and clarity is good.

2. The numerical proposal is sound.

3. The contribution of this work is hard to gauge due to point 3,4,5 in the weakness section. So the result, while looking promising, is currently only a minor strength of this work.

**Weaknesses:**

1. The authors omitted connections to constrained optimal transport, such as Benamou et al 2014 (1412.5154) and Tang et al 2024 (2403.05054), which are both general-purpose algorithms with entropic regularization.

2. There are numerous instances of grammatical errors in this work. The authors are advised to change it. (This doesn't affect this reviewer's evaluation for the score.)

3. While OFT-Sinkhorn with edge/node/node&edge constraints is definitely a good contribution, it doesn't seem clear why one would not simply use Sinkhorn in the unconstrained case. The authors have not included numerical comparison to make this point across. What is better? Sinkhorn has more GPU acceleration and should be much better in the unconstrained case (say, you take the same regularization strength and compare results).

4. In the numerical experiments, the proposed OFT-Sinkhorn algorithm seems to do better than Gurobi, which is quite confusing as Gurobi should output the ground truth. The reviewer suspects that it is because the OFT-Sinkhorn solution is unfeasible, thereby giving a better objective. If this interpretation is correct, then the authors need to change to a better metric for comparison.

5. The benchmark ZKW and Real methods are exact methods and they do not seem to be up to date. The authors need to also consider state-of-the-art methods for comparison, and the authors need to consider approximate algorithms.

6. The conclusion section needs a rewrite. It seems to suggest that this method can solve NP-hard graph-based problems, which is why the wording needs to be refined.

**Questions:**

No questions. All are suggestions listed in weakness already.

---

> ### Author Response · Authors · 2024-11-20
> **Part one**
>
> Thank you for your valuable feedback. We will adress all the grammatical errors  in the next version. Hopefully, our response below adequately addresses your concerns.
>
> ### Q1.The authors omitted connections to constrained optimal transport
> Thank you for your suggestion, we will provide a related description:
>
> Both [1, 2] and our method are based on entropic regularized algorithms and use the objective as an evaluation metric.
>
> The main difference is that [1, 2] are bipartite graph-based algorithms and the constraints used  are different.
>
> [1] also mentions transportation problems with capacity constraints:
> \begin{equation}
> \min_{\mathbf{P} \in \mathbb{R}_+^{N \times N}} \{ \langle \mathbf{D}, \mathbf{P} \rangle - \epsilon H(\mathbf{P}) \; ; \; \mathbf{P} \mathbf{1} = \mathbf{p}, \; \mathbf{P}^T \mathbf{1} = \mathbf{q}, \; \mathbf{P} \leq \mathbf{S} }
> \end{equation}
>
> The difference lies in:
> **1.** We extend OT on transport problem to a more general graph case (the MCF problem).
>
> **2.** The capacity  constraints in [1] are limited on **bipartite graphs**, while ours can impose capacity on any node or edge.
>
> **3.** [1] follows marginal constraint while we use flow balance constraint:
> \begin{equation}
> \min_{\mathbf{P} \in \mathbb{R}_+^{N \times N}} { \langle \mathbf{D}, \mathbf{P} \rangle - \epsilon H(\mathbf{P}) \; ; \; \mathbf{P} \mathbf{1} = \mathbf{q+d}, \; \mathbf{P}^T \mathbf{1} = \mathbf{q-s+d}, \; \mathbf{P} \leq \mathbf{S} }
> \end{equation}
> To the best of our knowledge, we are the first to apply entropic regularization to solve approximate solutions to the MCF problem with capacity constraint.
>
>
> [2] primarily consisting of inequality constraints. However, the formulations differ: in [2], the inequality constraint is on the assignment matrix, while still retaining the original Sinkhorn-style marginal  constraints:
>
> \begin{equation}
> \min_{\mathbf{P} \in \mathbb{R}_+^{N \times N}} { \langle \mathbf{D}, \mathbf{P} \rangle + \epsilon H(\mathbf{P},s_1, \ldots, s_K) \; ; \; \mathbf{P} \mathbf{1} = \mathbf{p}, \; \mathbf{P}^T \mathbf{1} = \mathbf{q} \;  }
> \end{equation}
>
> Whereas our work transforms the marginal probability constraints into flow balance constraints and provide a new perspective to introdece new iteration variable **q**, for details please refer to Appendix-C.1.
>
>
> Those works are parallel relationship and can be complementary. In the future, we could consider combining their constraints to solve new problems.
>
>
> ### Q2 It doesn't seem clear why one would not simply use Sinkhorn in the unconstrained case.
> Apologies for any misunderstanding. We would like to provide the following clarification and hope it addresses your concerns.
> The original Sinkhorn algorithm [1] is a matrix iterative algorithm applied to transportation problems, where an entropic regularization term is introduced to obtain an approximate solution. Our algorithm extends the original Sinkhorn from bipartite graphs to more general graph cases.  The flow balance constraint of ours is :
> $$
> \mathbf{P}\mathbf{1}_n - \mathbf{P}^\top \mathbf{1}_m = \mathbf{s}$$
> It is different at the optimization/formulation level. The formulation of the graph we focus on is different from previous work. Therefore, even in unconstrained scenarios, the traditional Sinkhorn cannot solve problems with flow balance.
> They are unable to handle transportation problems when: 1) the marginals are unknown  2) the presence of isolated points in the graph.
>
> ### Q3  The proposed OFT-Sinkhorn algorithm seems to do better than Gurobi, which is quite confusing.
>
>
> Sorry, there may have some misunderstanding here. We hope following clarification can resolve your concerns. Gurobi is a classic commercial solver for combinatorial optimization problems, and it is indeed always capable of finding the optimal solution. However, Gurobi is not efficient and cannot be run in parallel on GPUs.
> Here is partial result from Table 1:
>
> |Size|Methods|obj (W/o constraint)|time (W/o constraint)|
> |-|-|-|-|
> | 500 x 500| Real|10.02|14.8 s|
> || ZKW| 10.02 | 18.4 s|
> || Gurobi| 10.02| 768 s|
> || ours| 10.11| **4.18 s**|
> | 5k x 5k|Real|10.059|1843 s|
> ||ZKW| 10.059|2034 s|
> || Gurobi|-|≥ 3 hours|
> || ours|10.311|**33 s**|
> |10k x 10k|Real|10.08|3363 s|
> ||ZKW|10.08|3781 s|
> ||Gurobi|-|≥ 3 hours|
> ||ours|10.728|**47 s**|
>
> From the table above, it can be seen that for 500 nodes (batch_size=64), Gurobi's objective value (obj) is indeed the ground truth, but our algorithm is significantly faster. When the number of nodes exceeds 5k (batch_size=16), Gurobi's solving speed becomes too slow, so we set a time limit for computation. Gurobi exceeded this limit, and the result shown here is its intermediate solution. However, with enough time, it is indeed possible to obtain the optimal solution.

---

> > ### Author Response · Authors · 2024-11-20
> > **Part two**
> >
> > ### Q4 The benchmark ZKW and Real methods are exact methods and they do not seem to be up to date.
> >
> > Thank you for the reminder. We are currently testing the baseline of the latest methods and will report the results shortly. Additionally, our method's advantage lies in a matrix iteration-based approximation algorithm that can be used to design neural networks. Just as Sinkhorn can serve as an output layer in neural network to output doubly stochastic matrices [4], our algorithm can also be used to meet flow balance constraints in various machine learning models [5], offering huge potential for further exploration.
> >
> >
> > ### Q5 The conclusion section needs a rewrite.
> > Thank you for your valuable feedback. We will make the  revisions in the new version. Currently, our algorithm does not solve **NP** hard problems, but recent work has attempted to apply optimal transport based algorithms to solve **NP** problems [6]. Since our algorithm is closely related to flow, we plan to explore flow problems on graphs in the future, such as the **maximum minimum-cost flow** problem.
> > ### Reference
> > [1] Iterative Bregman projections for regularized transportation problems. SIAM 2015
> >
> > [2] A Sinkhorn-type Algorithm for Constrained Optimal Transport. arXiv 2024
> >
> > [3] The Sinkhorn–Knopp algorithm: convergence and applications. SIMAX 2008
> >
> > [4] Learning combinatorial embedding networks for deep graph matching. CVPR 2019
> >
> > [5] Gflownet foundations. JMLR 2023
> >
> > [6]Group Sparse Optimal Transport for Sparse Process Flexibility Design. IJCAI 2023

---

> > > ### Comment · Reviewer_T4Wu · 2024-11-20
> > > **Reply to the author**
> > >
> > > The reviewer apologizes for weakness 4, as the work does show that Gurobi has a better objective compared to Sinkhorn.
> > >
> > > The minor misunderstanding aside, the reviewer thinks that the lack of benchmarks for approximate algorithms is still a major red flag. While the reviewer appreciates the detailed comments posted, the reviewer still thinks this work does not fully justify the need to use entropic regularization compared to existing methods. The difference in performance between "Real" and the proposed method is still quite small. The reviewer suspects that the gap in runtime would disappear once one asks the OFT-Sinkhorn algorithm to have higher accuracy in the objective function. Furthermore, it is frankly quite baffling that the numerical performance doesn't obey the O(n^2) scaling law that would be suggested by the Sinkhorn-like algorithm formulation. So the reviewer can only conclude that the iteration number for larger instances is tweaked to be smaller to give a smaller runtime complexity, or the hardware has unexplained behavior that makes the scaling not behave according to theoretical complexity. To strengthen the manuscript, the reviewer suggests that the author add more existing benchmarks within the flow network community.

---

> > > > ### Author Response · Authors · 2024-11-22
> > > > **Part 1**
> > > >
> > > > Thank you for your valuable feedback. We hope our response further clarifies your concerns.
> > > > ### Q1  This work does not fully justify the need to use entropic regularization compared to existing methods.
> > > > Sure, we will further elaborate on the main contributions of our entropy regularization approach. Although entropy regularization has been used extensively in the optimal transport problem, yielding numerous results, our contribution builds on and extends these prior efforts.
> > > >
> > > > The idea of using entropy to regularize the optimal transport problem dates back to modeling concepts in transportation theory [Wilson, 1969]. In practice, actual traffic patterns in a network do not match those predicted by solutions to the optimal transport problem. **Real traffic** tends to be more diffuse, unlike the concentrated solutions of traditional optimal transport, which rely on only a few routes due to the sparsity of optimal couplings.
> > > >
> > > > Admittedly, in certain cases, our algorithm may not achieve state-of-the-art performance. Nevertheless, both our algorithm and theoretical insights hold significant value within the flow community, as highlighted by the following points:
> > > >
> > > > **1. Alignment with Real-World Transport Scenarios:** Entropy regularization better reflects real-world transport dynamics [1, 2]. Actual traffic patterns differ from those suggested by optimal transport solutions, as they tend to be more distributed. Our approach models such scenarios more effectively, especially considering the **MCF** problem, which generalizes the transport problem.
> > > >
> > > > **2. Efficiency through Matrix Iteration and GPU Utilization:** A major advantage of entropy regularization is its formulation as a matrix iteration problem, which allows us to leverage GPU acceleration. We are currently the only approach capable of applying entropic regularization with matrix iteration to the MCF problem.
> > > >
> > > > **3. Extension to General Graphs:** Compared to previous entropic OT methods, we have extended the entropic regularization algorithm from bipartite graphs to general graphs. This represents a significant theoretical expansion and supplementation of the original method.
> > > >
> > > >
> > > > ### Q2 The difference in performance between "Real" and the proposed method is still quite small.
> > > >
> > > > We apologize for the misunderstanding, and we will clarify the experimental setup more thoroughly in the next version of the paper. Our algorithm is more effective than Real at solving **dense graphs** problems.
> > > > In **Table 1**, under the 10k × 10k W/edge+node constraint, Real is indeed slightly faster than our algorithm. However, across other scales and constraints, our algorithm outperforms Real. Furthermore, under integer precision (as shown in **Table 2**), our algorithm is consistently faster than Real.
> > > >
> > > > We believe the main factor here is the sparsity of the graph. As shown in Appendix D, the out-degree of the Uniform dataset is typically 8, making it a **sparse** graph, whereas the Integer dataset usually has an edge count exceeding  **nlog n**, making it a dense graph. Matrix-iteration-based algorithms tend to perform better on **dense** graphs. If needed, we would be happy to provide more detailed decoupled experiments on graph density for further analysis.

---

> > > > > ### Author Response · Authors · 2024-11-22
> > > > > **Part 2**
> > > > >
> > > > > ### Q3 The numerical performance doesn't obey the O(n^2) scaling law. The iteration number for larger instances is tweaked to be smaller to give a smaller runtime complexity
> > > > >
> > > > > Thank you for your valuable feedback. In contrast to the original Sinkhorn iterations, our approach introduces iterations involving the calculation of a new variable **q**, which makes our procedure  more complex. The previous **O(n^2)** scaling law pertains specifically to the original Sinkhorn complexity and does not  reflect our setting.
> > > > >
> > > > > Furthermore, in our large-instance experiments, we did tweak the iteration number. We maintained consistent convergence criteria across all experiments.
> > > > >
> > > > > Compared to the standard optimal transport problem, MCF involves a more complex setting, which inevitably impacts performance. Therefore, we believe that the **density of the graph** plays a crucial role. Unlike the traditional optimal transport problem, which typically assumes a fully connected bipartite graph with a given distance matrix, our general graph involves an additional connectivity density metric. We have supplemented the results to include experiments on dense and fully connected graphs as follows:
> > > > >
> > > > >
> > > > > |Instance/Avg Outdegree|8|16|32|
> > > > > |-|-|-|-|
> > > > > |10k\_Real|10.04(1424 s)|9.99(9302 s)|9.97(42740 s)|
> > > > > |10k\_W/EN(Ours)|10.29(990 s)|10.18(1744 s)|10.06(3646 s)|
> > > > >
> > > > >
> > > > > It can be seen that as the density of the graph increases, the advantages of the matrix iteration algorithm become increasingly apparent.
> > > > >
> > > > > ### Q4 Add more existing benchmarks within the flow network community.
> > > > > Thank you for your suggestions. We will expedite our efforts and provide more comparative experiments before the rebuttal deadline.
> > > > >
> > > > > ### Reference
> > > > > [1] Information geometry connecting Wasserstein distance and Kullback–Leibler divergence via the entropy-relaxed transportation problem. Information Geometry. 2018
> > > > >
> > > > > [2] Sinkhorn distances: Lightspeed computation of optimal transport. NIPS 2013

---

> ### Author Response · Authors · 2024-11-28
>
> ### More comparison with SOTA/Approximate MCF Method
> Here, we  further incorporated the suggestions of the reviewers by adding more comparison between our method and SOTA MCF algorithm. We observed that their experiments focused on **extremely large sparse graphs**. To align with this, we've re-implemented our algorithm to better suit the triplet structure of sparse graphs. For a comparative analysis, we used the three massive datasets they published, pitting our method against theirs. Additionally, we ran their approach on the medium and small datasets from Table 2 in our paper, with the results summarized as follows.
> The **pns** method refers to the approach outlined in [1], while **lemon**[2] is a classic solver for MCF problems(Here we test on lemon-ns which is the network simplex version of lemon). The new dataset(Netgen_8_20/Netgen_lo_8_20/Vision_inv_05) used in this study are all sourced from [1]. Besides the original generater(NETGEN) taken in our main paper, Vsion instance is  based on large-scale maximum flow problems arising in computer vision applications.The EOFT timings were tested on an NVIDIA GeForce RTX 4090 GPU.
>
> |**Methods** | **Netgen_100 Obj**|**Netgen_100 Time** | **Netgen_500 Obj** | **Netgen_500 Time**|**Netgen_1000 Obj** |**Netgen_1000 Time**|**Netgen_5000 Obj** |**Netgen_5000 Time**| **Netgen_8_20 Obj**|**Netgen_8_20 Time** |  **Netgen_lo_8_20 Obj** | **Netgen_lo_8_20 Time** |**Vision_inv_05 Obj** |**Vision_inv_05 Time** |
> | -|-|-|-| -|-|-|-| -|-|-|-|-|-|-|
> |Real | 64.58 | 421 ms | 20.418 | 1028 s | 23.36 | 2340 s |15.65 | 18592 s|-- | $\geq{24}$ hours | -- | $\geq{24}$ hours | -- | $\geq{24}$ hours |
> |ZKW | 64.58 | 504 ms | 20.418 | 706 s | 23.36 | 1639 s|15.65 | 7491 s |-- | $\geq{24}$ hours | -- | $\geq{24}$ hours | -- | $\geq{24}$ hours |
> |Gurobi | 64.583 | 77.3 s | 20.42 | 4572 s | -- | $\geq{10}$ hours | -- | $\geq{10}$ hours | -- | $\geq{24}$ hours | -- | $\geq{24}$ hours |
> |LEMON-ns | 64.582  | 56 ms  | 20.43 | 7.2 s | 23.57 | 6.6 s|15.68 |41 s| 18.33|186 s |12.88|51 s|30.16|2805 s
> |pns(k=1,p=1) | 64.582 | 469 s | 20.44 | 33941 s | 23.56 | 28081 s | 15.66 | 26718 s | 18.33 | 335 s | 12.88 | 89 s |30.16| 1774 s |
> |pns(k=4,p=4) | 64.582| 303  s   | 20.44 | 20451 s | 23.56 | 13301 s| 15.66| 10231 s  | 18.33| 261 s | 12.88 | 261 s| 30.16| 2579 s |
> |pns(k=16,p=16) | 64.582 | 256 s | 20.44 | 15388 s | 23.56 | 10117 s |  15.66 | 67847 s | 18.33 | 286 s | 12.88| 286 s |30.16| 2512 |
> |**EOFT($\epsilon-big$)** | 7981 | 0.3 s | 506027 | 0.6 s | 793392 | 8.6 s|43182 | 8.6 s | 485.1 | 2.97 s | 218.07 | 1.16 s | 632.1 | 19.5 s  |
> |**EOFT($\epsilon-mid$)** | 103.57 | 1.36 s | 5009 | 2.8 s | 6341 | 13.5 s|19.53 |20.5 s | 19.44| 8.01 s | 78.46 | 4.3 s |820421019 | 278 s  | |
> |**EOFT($\epsilon-small$)** | 65.05 | 18.2 s | 20.548 | 82 s | 24.696 | 420 s |16.41 |1261 s | 18.83 | 50.4 s | 13.08 | 59.3 s | 32.04 |1802 s|
>
> This table is the experiment setting:
> | **Instance** | **Instance Num**   | **Node Num** | **Arc Num**  |**Total Supply** | **Capacity_Range**| **Cost_Range** | **err**    |**ε Range** |**d0**|
> |-|-|-|-|-|-|-|-|-|-|
> | **Netgen_8_20** |1  | 1048576| 8388608 |1024000| [1,1000]    | [1,10000]  | 1e-3 |[0.01，0.1, 1]|1e-4|
> | **Netgen_lo_8_20** |1 |1048576  | 8388608|10240 | [1,1000]  | [1,10000]  | 1e-3  | [0.01，0.1, 1]|1e-4|
> | **Vision_inv_05** |1   |3899394  | 23091149 |10000 |  [0,100]    | $\frac{[9000,11000]}{capacity}$ |1e-3 |[0.01，0.1, 1]  |1e-4|
> | **Netgen_100** |128|100 |800|1000| [500,1000]    | [10,100]    |1e-4  |[5e-4,5e-3,5e-2]     |1e-4|
> | **Netgen_500**  |256|500  | 64000  | 1000 |   [500,1000]    |[10,100]  |1e-4 |[5e-4,5e-3,5e-2]|1e-4|
> | **Netgen_1000** |128|1000 | 80000 |1000 |   [500,1000]    | [10,100] |1e-4 |[5e-4,5e-3,5e-2]|1e-4|
> | **Netgen_5000**  |8  |5000 | 1000000| 1000 |  [500,1000] | [10,100] |1e-4 |[5e-4,5e-3,5e-2]|1e-4|
>
> From the results, we have found that Lemon is best suited for running on medium and small datasets, as it is a highly optimized CPU-based MCF solver. On the three newly added large-scale datasets, Real/ZKW/Gurobi exceeded the maximum time limit we set, hence represented as ‘-’, while our algorithm can quickly obtain a stable approximate-optimal solution.
>
> **As the first work to apply entropic regularization to the MCF problem and achieve a GPU-friendly algorithm**, our algorithm effectively addresses the minimum cost flow problem on extremely large sparse graphs, and in some instances, even outperforms the current **SOTA** algorithms in obtaining approximate solutions. This demonstrates the huge potential of matrix-iteration-based algorithms for designing more efficient GPU algorithms in the **flow community**. Hope that our additional experiments will enhance your confidence and recognition of our work.
> ### Reference
> [1] Parallel network simplex algorithm for the minimum cost flow problem. Concurrency and Computation: Practice and Experience, 2022
> [2] Efficient implementations of minimum-cost flow algorithms. arxiv 2012

---

### Official Review · Reviewer_6h6u · 2024-10-29

**Soundness:** 3
**Presentation:** 3
**Contribution:** 2
**Rating:** 6
**Confidence:** 3

**Summary:**

This paper introduces the concept of Optimal Flow Transport (OFT) as an extension of vanilla OT, where the authors consider a more general graph setting and the marginal constraints in vanilla OT are replaced by flow balance constraints. Adding entropy regularization directly cannot derive an algorithm to obtain an approximate solution as directly as vanilla OT due to two reasons. First of all, there are many isolated points that do not have any flow passing through them in the exact solution, while under entropy regularization, flows are necessarily channeled through them, causing significant bias. Second, the flow balance constraint cannot directly form an alternating iterative algorithm like the marginal constraint. The authors address these issues by modifying the original problem formulation and proposing the new OFT-Sinkhorn algorithm. The new approach utilizes GPU-friendly matrix iterations and is guaranteed to converge globally (with the rate). The authors also extend their approach to a more general setting where the capacity constraints on nodes
and edges are avaliable. The encouraging empirical results are presented and discussed.

**Strengths:**

1. The authors extend the vanilla OT within a bipartite graph to a more general graph case, and thus propose optimal flow transport, in which the marginal constraints are replaced by flow balance constraints.

2. The authors propose the entropic OFT to get the GPU-friendly OFT-Sinkhorn algorithm to get the approximate solution of the OFT problem. The theoretical guarantee is also proposed for global convergence.

3. The authors incorporate node and edge capacity constraints into the OFT, and in this case our OFT is equivalent to the minimum-cost flow problem. By considering these constraints, the authors modify the OFT-Sinkhorn algorithm to ensure that the output satisfies capacity constraints. Experimental results on the minimum-cost flow problem showcase the superiority of the proposed algorithm.

**Weaknesses:**

1. The idea of extending the vanilla OT within a bipartite graph to a more general graph case is not new and have been investigated in the literature. While I agree that it is meaningful to replace the marginal constraints by by flow balance constraints on nodes, this idea is novel by itself and the optimal flow transport problem in Eq. (2) is rather standard.

2. It is important to note that directly adding entropy regularization cannot derive an algorithm to obtain an approximate solution as directly as vanilla OT. However, I do not think addressing this issue would be difficult and the modified formulation is Eq. (5) is also based on some standard tricks from network flow literature. Based on this formulation, the development of OFT-Sinkhorn algorithm is almost the same as that of Sinkhorn algorithm and handling node and edge capacity constraints would not significantly change the algorithmic scheme.

I would like to say that I do not undermine the value of new methods but the contribution of this paper is a little bit incremental to me from a technical viewpoint. This is a good work by itself but does not offer too much insights.

**Questions:**

1. I encourage the authors to further clarify the difference between the current paper and the existing works. For example, you state that the previous studies primarily rely on graph’s shortest distances and do not directly compute transport couplings within the graph. In my understanding, these studies can compute transport couplings but need to precompute the shortest distances on the graph (please correct me if I am wrong). Then, why can you directly compute transport couplings within the graph? It seems counterintuitive to me if this is only because you use the entropy regularization. Would you like to provide me more details?

2. You state that your EOFT-Sinkhorn is evaluated on high-precision and integer-precision MCF problems with different scales from one hundred to five thousand size, exhibiting significant time efficiency and the ability to approximate optimal solutions. The experimental results confirm that your method can achieve a solution of higher quality. It is a bit surprising since the vanilla implementation of Sinkhorn algorithm requires much time to converge to a high-accurate solution. Would you like to give me some reasons why your method could achieve the high-accurate solutions in practice?

---

> ### Author Response · Authors · 2024-11-20
>
> ### Q1 Further clarify the difference between the current paper and the existing works
>
> #### Q1.1  In my understanding, these studies can compute transport couplings but need to precompute the shortest distances on the graph.
> Sorry for any misunderstanding. Yes, they can compute transport couplings, but they first need to calculate the shortest path before solving the optimal transport problem between the source and target nodes.
>
> As demonstrated in [1, 2], their algorithms do not directly compute the flow and are unable to impose **capacity constraints** on the nodes. Consequently, they cannot address **constrained MCF** problems as our method does.
>
> Additionally, they are not **GPU-friendly**  and cannot be embedded into neural networks. They will cause gradient truncation while ours and Sinkhorn[3] allowing the gradient backpropagation.
>
> #### Q1.2  Why can you directly compute transport couplings within the graph?
> We will explain why our method can directly compute the coupling from three perspectives,  hope this will address your concerns.
>
> **1.** Compared to [4], we address the iteration issue for **isolated points** on the graph, a problem not considered in previous works. By introducing virtual transport flows from each node to itself, we enable isolated points to participate in the iterations, even if they originally have no incoming or outgoing flows.
>
> **2.**  We extend the **marginal probability constraints** in the original Sinkhorn algorithm:
> $$\mathbf{P}\mathbf{1} = \mathbf{a}, \mathbf{P}^\top\mathbf{1}=\mathbf{b}$$
> into **flow balance constraint**:
> $$
> \mathbf{P}\mathbf{1}_n - \mathbf{P}^\top \mathbf{1}_m = \mathbf{s}$$
>
> aligned with the previous Sinkhorn formulation, enabling us to adopt a similar entropic regularization approach to develop a matrix-iteration-based algorithm. As a result, just as the original Sinkhorn algorithm computes the coupling for transportation problems, our method computes the coupling for **MCF** problems.
>
> **3.** We introduced a new marginal variable **q**, and no previous work has derived the iteration formula as we did. The detailed derivation can be found in Appendix-C.1.
> ### Q2 Would you like to give me some reasons why your method could achieve the high-accurate solutions in practice?
> Our results are actually similar to Sinkhorn. In matrix-iteration algorithms, there is a trade-off between accuracy and time: the higher the accuracy (i.e., the smaller the $\epsilon$), the more iterations are required, leading to longer computation times. Both our method and Sinkhorn exhibit this characteristic.
>
> In our experiments, we set **\epsilon** between $1 \times 10^{-3}$ and $1 \times 10^{-4}$, achieving a solution within 5% of the optimal. Compared to other methods, our **matrix-based** algorithm is more efficient. **Real** and **ZKW**, based on augmentation paths, are slower on dense graphs, while Gurobi becomes too slow on larger-scale problems.
>
> Below are decoupled experiments showing the relationship between $\epsilon$ and the number of iterations:
> | Instance/$\epsilon$|1|0.1|0.01|1e-3|5e-4|
> |-|-|-|-|-|-|
> |500\_W/o(Ours)|9|21|405|503|662|
> |500\_W/EN(Ours)|26|209|262|13036|24838|
> |500\_sinkhorn|11|101|3601|NAN|NAN|
> |500\_sinkhorn(log)|17|131|3781|37571|89724|
> The constraint settings here are consistent with Table 1 in  main paper. As $\epsilon$ decreases, both our method and Sinkhorn experience an increase in the number of iterations and a corresponding increase in computation time.
> ### Q3 The idea of extending the vanilla OT within a bipartite graph to a more general graph case is not new.
> Thank you for your valuable feedback.
> One of the main contribution and potential of our algorithm lies in transforming the original OT marginal probability constraints P_1 = a, P_1^T = b into flow balance constraints P_1 - P_1^T = s. This transformation allows our algorithm to satisfy various constraints, such as the capacity and flow balance constraints in the MCF problem or the flow constraints in TSP problems:
> \begin{equation}
> % \textstyle
> \min_{\mathbf{X} \in U(\mathbf{1}, \mathbf{1})} \langle \mathbf{D}, \mathbf{X} \rangle \quad \text{s.t.} \, \mathbf{P} \leq (n-1)\mathbf{X}, \, \mathbf{P}\mathbf{1} - \mathbf{P}^\top \mathbf{1} = \mathbf{s}
> \end{equation}
> The entropic regularization method has enabled us to develop a GPU-friendly matrix iterative algorithm, which can be embedded into the output layer of neural networks (similar to the original Sinkhorn layer [3]).
> ### Reference
> [1] Sobolev transport: A scalable metric for probability measures with graph metrics. AISTATS 2022
>
> [2] Generalized Sobolev Transport for Probability Measures on a Graph. ICML 2024
>
> [3] Learning combinatorial embedding networks for deep graph matching.CVPR 2019
>
> [4] The Sinkhorn–Knopp algorithm: convergence and applications. SIMAX 2008
>
> [5]Pot: Python optimal transport
>
> [6] Gflownet foundations. JMLR 2023
>
> [7] Stabilized sparse scaling algorithms for entropy regularized transport problems. SISC 2019

---

> ### Author Response · Authors · 2024-11-28
>
> May we ask if we have addressed your concern? We do hope we could resolve your any potential questions!

---

### Official Review · Reviewer_42rw · 2024-11-03

**Soundness:** 3
**Presentation:** 3
**Contribution:** 2
**Rating:** 6
**Confidence:** 2

**Summary:**

The authors provide a Sinkhorn-like algorithm (hence involving parallelizable matrix multiplication) for the case of optimal transport on more general graphs than the bipartite graph.

**Strengths:**

The paper is well written and theoretically sound and possibly introduces a new interesting method that can leverage modern GPU infrastructure.

**Weaknesses:**

The numerical experiments are not completely clear, and their relation to existing works could be slightly clarified.

**Questions:**

In Table 1, there is an error in the 10kx10k line for the W/ edge+note constraints: the fastest method in 'Real" with 1425s not "ours" with 1462s.

In general, when there are constraints, the "Real" method has similar orders of magnitude results and still has a better solution quality in terms of objective. So what if the authors stop the compared algorithms at the same obj value their algorithm achieves? It is unclear which algorithm would be faster between "Real" and "ours"?

"However, these studies primarily rely on graph's shortest distances and do not directly compute transport couplings within the graph." -> How is that a problem? Is it slower? Not GPU friendly? Or is it not as generic as the authors' solution? Are there settings where the authors could compare their approach with theirs?

---

> ### Author Response · Authors · 2024-11-20
>
> We would like to express our sincere gratitude for thoroughly evaluating our paper and providing insightful and valuable feedback. We are genuinely committed to addressing your concerns and respond to your specific comments below.
>
> ### Q1. There is an error in the 10kx10k line for the W/ edge+note constraints.
> Sorry for the writing errors，we  fixed it in the next version main paper,thank you for your valuable advice.
>
> ### Q2.What if the authors stop the compared algorithms at the same obj value their algorithm achieves?
> Thank you for the reminder. However, these algorithms do not support using the objective (**Obj**) value as an early stopping criterion.
>
> Both **Real and ZKW** are augmentation path-based algorithms that repeatedly check if the residual network can accommodate more flow. As a result, their **Obj** value increases throughout the solving process, making it unsuitable for early termination.
>
> Regarding **Gurobi**, experiments show that when solving MCF problems, it uses either the dual simplex or primal simplex algorithm. During iterations, the objective value only change between **Inf** and the optimal solution (e.g., in an experiment with 500 nodes, the first 10,089 iterations showed "Inf," and the optimal solution was reached only after the 10,090th iteration). This behavior likely stems from the nature of the MCF problem.
>
> In contrast, our **matrix-based approximation** algorithm can quickly provide approximate solutions, offering a significant speed advantage.
>
> ### Q3 It is unclear which algorithm would be faster between "Real" and "ours"?
> We apologize for the misunderstanding, and we will clarify the experimental setup more thoroughly in the next version of the paper. Our algorithm is more effective than Real at solving **dense graphs** problems.
> In **Table 1**, under the 10k × 10k W/edge+node constraint, Real is indeed slightly faster than our algorithm. However, across other scales and constraints, our algorithm outperforms Real. Furthermore, under integer precision (as shown in **Table 2**), our algorithm is consistently faster than Real.
> We believe the main factor here is the sparsity of the graph. As shown in Appendix D, the out-degree of the Uniform dataset is typically 8, making it a **sparse** graph, whereas the Integer dataset usually has an edge count exceeding  **nlog n**, making it a dense graph. Matrix-iteration-based algorithms tend to perform better on **dense** graphs. If needed, we would be happy to provide more detailed decoupled experiments on graph density for further analysis.
>
> ### Q4 The explanation for "these studies primarily rely on graph's shortest distances" Is it slower?
> Apologies for not clearly explaining our advantages. While OT algorithms on graphs may not be slower—and could even be faster than ours—they are not **generic** or **GPU-friendly**.
>
> As shown in [1, 2], these algorithms do not compute the flow at intermediate nodes or enforce capacity constraints, so they cannot solve constrained MCF problems like ours. They also aren't compatible with neural networks and may cause gradient truncation, unlike our method and Sinkhorn[3], which support gradient backpropagation.
>
> These algorithms treat the graph problem as a transportation problem on the nodes, first calculating shortest paths before solving the optimal transport between source and target. They can't handle capacity constraints at nodes. In contrast, our matrix-based entropic regularization algorithm directly iterates to compute the coupling on the constrained graph.
>
> There is no  available PyTorch or Python code for comparison, but we will make every effort to replicate it and report it as soon as possible.
>
> ### Q5.Their relation to existing works could be slightly clarified.
> Thank you for your valuable advice. We will provide a more detailed explanation of the settings on our next  main papaer. Next, we will explain the differences between our method and other approaches:
>
> As a more general form of the transportation problem, the **MCF** problem has been well studied. However, to the best of our knowledge, we are the first to apply **entropic regularization** for approximate solutions to the **MCF** problem. Given the use of entropic regularized methods in **machine learning**, they can be embedded in the output layer of neural networks to enable **gradient backpropagation**. Thus, the matrix-vector iterative algorithm we propose also holds potential for applications in fields like GFlowNets[4], where the generation process aligns with our algorithm's flow balance constraints, making it a promising tool for future GFlowNet designs.
>
> ### Reference
> [1] Sobolev transport: A scalable metric for probability measures with graph metrics. AISTATS 2022
>
> [2] Generalized Sobolev Transport for Probability Measures on a Graph. ICML 2024
>
> [3] Learning combinatorial embedding networks for deep graph matching. CVPR 2019.
>
> [4] Gflownet foundations. JMLR 2023

---

> ### Author Response · Authors · 2024-11-28
>
> May we ask if we have addressed your concern? We do hope we could resolve your any potential questions!

---

### Official Review · Reviewer_vSLu · 2024-11-03

**Soundness:** 3
**Presentation:** 2
**Contribution:** 2
**Rating:** 6
**Confidence:** 3

**Summary:**

This paper proposed efficient algorithms to address the minimum cost flow (MCF) problem with Sinkhorn iterations. The authors claim a problem named the Optimal Flow Transport (OFT) problem as an extension of OT to general graphs by replacing marginal constraints with flow balance constraints. They use entropic regularization and virtual flows to propose a GPU-friendly/matrix-based OFT-Sinkhorn algorithm. Furthermore, they apply capacity constraints on nodes and edges, transforming the OFT problem into a capacity-constrained MCF problem. Experiments demonstrate computational efficiency on various datasets.

**Strengths:**

This paper applied the entropic regularization and Sinkhorn iterations to the MCF problem and provided algorithms that can run on GPU. This is good for solving large flow problems more efficiently than the traditional method. The paper includes experimental evaluations demonstrating that the proposed algorithm can solve the problem efficiently on large-scale problems. In addition, this work provides the proof for the convergence of the proposed algorithms looks good to me.

**Weaknesses:**

My primary concern is about the novelty. This work claims to extend OT to general graphs with flow balance constraints, which is essentially a restatement of the classic minimum cost flow (MCF) problem. The MCF problem is well-studied, and applying entropic regularization is a common practice. Secondly, this paper needs further literature review of existing work that applies similar techniques to the MCF problem using GPU acceleration or matrix-based methods. such as [1][2]. Additionally, another concern is that this paper does not show how their proposed method benefits the machine learning community. There is a lack of discussion on the applications of MCF in machine learning tasks, and no such experiments are provided. Given these weaknesses, I tend to reject this paper.

[1] Harish, Pawan, Vibhav Vineet, and P. J. Narayanan. "Large graph algorithms for massively multithreaded architectures." International Institute of Information Technology Hyderabad, Tech. Rep. IIIT/TR/2009/74 (2009).

[2] Wu, Jiadong, Zhengyu He, and Bo Hong. "Efficient CUDA algorithms for the maximum network flow problem." GPU Computing Gems Jade Edition. Morgan Kaufmann, 2012. 55-66.

**Questions:**

- How does the proposed algorithm compare with existing GPU-accelerated methods for solving the MCF problem? Including such comparisons within the experiment will strengthen this paper.
- Is there any machine learning problem that can apply the algorithms of this paper?
- Have the authors evaluated the performance of their algorithm on CPUs? Will the code be public?

---

> ### Author Response · Authors · 2024-11-20
>
> Thank you for your thorough review. Hopefully, our response below adequately addresses your concerns.
>
> Sorry, we forgot to mention in the main paper that our code is provided in the supplementary files. We have added it in the next version.
> ### Q1. How does the proposed algorithm compare with existing GPU-accelerated  or matrix-based  methods.
>
> Thank you for the reminder. However, we are the only **GPU-accelerated** algorithm with a **matrix-based** approach. Other papers, such as [4, 5], focus on the **maximum flow problem**, which differs from the **minimum cost flow problem**. The maximum flow aims to maximize the flow without considering costs, while the MCF problem includes both capacity limits and associated costs.
> We reviewed existing MCF solvers, including [6, 7, 8, 9]. [6] offers a parallel network **simplex algorithm** but lacks open-source code. [7] presents the classic MCF solver **LEMON**, while [8] introduces a continuous shortest path algorithm, and [9] discusses a parallel simplex algorithm for CPU-based multi-threading. None of them can run on GPU.
> To our knowledge, we are the first to apply entropic regularization to the MCF problem and propose a **GPU-efficient** matrix iterative algorithm. If we've missed anything, we  would be happy to include additional comparative experiments.
>
> ### Q2. Is there any machine learning problem that can apply the algorithms of this paper?
> Thank you for your valuable advice. Our method can indeed be applied to machine learning, including solving NP-hard problems like the **TSP** and **traffic flow prediction**. Many real-world problems, such as the TSP, involve **flow balance constraints**, which can be formulated as:
>
> \begin{equation}
> % \textstyle
> \min_{\mathbf{X} \in U(\mathbf{1}, \mathbf{1})} \langle \mathbf{D}, \mathbf{X} \rangle \quad \text{s.t.} \, \mathbf{P} \leq (n-1)\mathbf{X}, \, \mathbf{P}\mathbf{1}_n - \mathbf{P}^\top\mathbf{1}_n = \mathbf{s}
> \end{equation}
>
> Here, $\mathbf{D}$ is the distance matrix, $\mathbf{X} \in U(\mathbf{1}, \mathbf{1})$ enforces $\mathbf{X}\mathbf{1} = \mathbf{1}$ and $\mathbf{X}^\top\mathbf{1} = \mathbf{1}$, $\mathbf{P}$ is the commodity flow matrix, and $\mathbf{s} = (n-1, -1, -1, \dots, -1)$ represents the flow starting from the first node.
>
> Our method satisfies these constraints and can be used as the output layer of a **neural network** to produce sparse assignment matrices, acting as supervision for the **TSP neural solver**.
>
> ### Q3. Have the authors evaluated the performance of their algorithm on CPUs?
> Thank you for the reminder. We will provide the CPU experiments of the algorithm as follows:
>
> |Size|Methods|time (W/o constraint)|time (W/ edge constraint)|time (W/ node constraint)|time (W/ edge+node constraints)|
> |-|-|-|-|-|-|
> | 500 x 500|CPUs|394 s|12118 s|4340 s|20248 s|
> |500 x 500|GPUs|4.2 s|34 s|43.4 s|72.8 s|
> |5k x 5k|CPUs|3230 s|78969 s|22265 s| 85166 s|
> |5k x 5k|GPUs|33 s|814 s| 316 s| 656 s|
>
>
> Here, we show a comparison of the algorithm's running time when executed on both CPU and GPU under four  constraints.
>
> ### Q4. The MCF problem is well-studied, and applying entropic regularization is a common practice.
> Although the **MCF** problems as well as the transportation problem is well-studied, but significant advances in optimal transport theory, such as entropic regularized algorithms for doubly stochastic matrices [1] or domain adaptation [2], continue to emerge. MCF problem is a more general form of the transport problem, and to our knowledge, we are the first to apply entropic regularization for approximate solutions to MCF.
>
> Given the application of these entropic regularized methods in the field of machine learning, they can be embedded in the output layer of neural networks  allowing backpropagation of gradients. Our matrix-based iterative algorithm also shows potential for applications in fields like GFlowNets[3], where the generation process aligns with our algorithm’s flow balance constraints, making it a promising tool for future GFlowNet designs.
> ### Reference
> [1] Learning combinatorial embedding networks for deep graph matching.CVPR 2019.
>
> [2] Optimal transport for multi-source domain adaptation under target shift.PMLR 2019
>
> [3] Gflownet foundations. JMLR 2023
>
> [4] Large graph algorithms for massively multithreaded architectures. IIIT Hyderabad 2009
>
> [5] Efficient CUDA algorithms for the maximum network flow problem. GPU Computing Gems Jade Edition 2012
>
> [6] Multi-granularity hybrid parallel network simplex algorithm for minimum-cost flow problems. J Supercomput 2020
>
> [7] Efficient implementations of minimum-cost flow algorithms. arXiv 2012
>
> [8] muSSP: Efficient Min-cost Flow Algorithm for Multi-object Tracking. NIPS 2020
>
> [9] Parallel network simplex algorithm for the minimum cost flow problem. CONCURR COMP-PRACT E 2022

---

> > ### Author Response · Authors · 2024-11-26
> >
> > Shall we ask that do you have any further concerns and we do hope we could resolve your potential questions!  We hope that our further answers can increase your confidence on our work.

---

> ### Author Response · Authors · 2024-11-28
>
> ### More comparison with SOTA/Approximate MCF Method
> Here, we  further incorporated the suggestions of the reviewers by adding more comparison between our method and SOTA MCF algorithm. We observed that their experiments focused on **extremely large sparse graphs**. To align with this, we've re-implemented our algorithm to better suit the triplet structure of sparse graphs. For a comparative analysis, we used the three massive datasets they published, pitting our method against theirs. Additionally, we ran their approach on the medium and small datasets from Table 2 in our paper, with the results summarized as follows.
> The **pns** method refers to the approach outlined in [1], while **lemon**[2] is a classic solver for MCF problems(Here we test on lemon-ns which is the network simplex version of lemon). The new dataset(Netgen_8_20/Netgen_lo_8_20/Vision_inv_05) used in this study are all sourced from [1]. Besides the original generater(NETGEN) taken in our main paper, Vsion instance is  based on large-scale maximum flow problems arising in computer vision applications.The EOFT timings were tested on an NVIDIA GeForce RTX 4090 GPU.
>
> |**Methods** | **Netgen_100 Obj**|**Netgen_100 Time** | **Netgen_500 Obj** | **Netgen_500 Time**|**Netgen_1000 Obj** |**Netgen_1000 Time**|**Netgen_5000 Obj** |**Netgen_5000 Time**| **Netgen_8_20 Obj**|**Netgen_8_20 Time** |  **Netgen_lo_8_20 Obj** | **Netgen_lo_8_20 Time** |**Vision_inv_05 Obj** |**Vision_inv_05 Time** |
> | -|-|-|-| -|-|-|-| -|-|-|-|-|-|-|
> |Real | 64.58 | 421 ms | 20.418 | 1028 s | 23.36 | 2340 s |15.65 | 18592 s|-- | $\geq{24}$ hours | -- | $\geq{24}$ hours | -- | $\geq{24}$ hours |
> |ZKW | 64.58 | 504 ms | 20.418 | 706 s | 23.36 | 1639 s|15.65 | 7491 s |-- | $\geq{24}$ hours | -- | $\geq{24}$ hours | -- | $\geq{24}$ hours |
> |Gurobi | 64.583 | 77.3 s | 20.42 | 4572 s | -- | $\geq{10}$ hours | -- | $\geq{10}$ hours | -- | $\geq{24}$ hours | -- | $\geq{24}$ hours |
> |LEMON-ns | 64.582  | 56 ms  | 20.43 | 7.2 s | 23.57 | 6.6 s|15.68 |41 s| 18.33|186 s |12.88|51 s|30.16|2805 s
> |pns(k=1,p=1) | 64.582 | 469 s | 20.44 | 33941 s | 23.56 | 28081 s | 15.66 | 26718 s | 18.33 | 335 s | 12.88 | 89 s |30.16| 1774 s |
> |pns(k=4,p=4) | 64.582| 303  s   | 20.44 | 20451 s | 23.56 | 13301 s| 15.66| 10231 s  | 18.33| 261 s | 12.88 | 261 s| 30.16| 2579 s |
> |pns(k=16,p=16) | 64.582 | 256 s | 20.44 | 15388 s | 23.56 | 10117 s |  15.66 | 67847 s | 18.33 | 286 s | 12.88| 286 s |30.16| 2512 |
> |**EOFT($\epsilon-big$)** | 7981 | 0.3 s | 506027 | 0.6 s | 793392 | 8.6 s|43182 | 8.6 s | 485.1 | 2.97 s | 218.07 | 1.16 s | 632.1 | 19.5 s  |
> |**EOFT($\epsilon-mid$)** | 103.57 | 1.36 s | 5009 | 2.8 s | 6341 | 13.5 s|19.53 |20.5 s | 19.44| 8.01 s | 78.46 | 4.3 s |820421019 | 278 s  | |
> |**EOFT($\epsilon-small$)** | 65.05 | 18.2 s | 20.548 | 82 s | 24.696 | 420 s |16.41 |1261 s | 18.83 | 50.4 s | 13.08 | 59.3 s | 32.04 |1802 s|
>
> This table is the experiment setting:
> | **Instance** | **Instance Num**   | **Node Num** | **Arc Num**  |**Total Supply** | **Capacity_Range**| **Cost_Range** | **err**    |**ε Range** |**d0**|
> |-|-|-|-|-|-|-|-|-|-|
> | **Netgen_8_20** |1  | 1048576| 8388608 |1024000| [1,1000]    | [1,10000]  | 1e-3 |[0.01，0.1, 1]|1e-4|
> | **Netgen_lo_8_20** |1 |1048576  | 8388608|10240 | [1,1000]  | [1,10000]  | 1e-3  | [0.01，0.1, 1]|1e-4|
> | **Vision_inv_05** |1   |3899394  | 23091149 |10000 |  [0,100]    | $\frac{[9000,11000]}{capacity}$ |1e-3 |[0.01，0.1, 1]  |1e-4|
> | **Netgen_100** |128|100 |800|1000| [500,1000]    | [10,100]    |1e-4  |[5e-4,5e-3,5e-2]     |1e-4|
> | **Netgen_500**  |256|500  | 64000  | 1000 |   [500,1000]    |[10,100]  |1e-4 |[5e-4,5e-3,5e-2]|1e-4|
> | **Netgen_1000** |128|1000 | 80000 |1000 |   [500,1000]    | [10,100] |1e-4 |[5e-4,5e-3,5e-2]|1e-4|
> | **Netgen_5000**  |8  |5000 | 1000000| 1000 |  [500,1000] | [10,100] |1e-4 |[5e-4,5e-3,5e-2]|1e-4|
>
> From the results, we have found that Lemon is best suited for running on medium and small datasets, as it is a highly optimized CPU-based MCF solver. On the three newly added large-scale datasets, Real/ZKW/Gurobi exceeded the maximum time limit we set, hence represented as ‘-’, while our algorithm can quickly obtain a stable approximate-optimal solution.
>
> **As the first work to apply entropic regularization to the MCF problem and achieve a GPU-friendly algorithm**, our algorithm effectively addresses the minimum cost flow problem on extremely large sparse graphs, and in some instances, even outperforms the current **SOTA** algorithms in obtaining approximate solutions. This demonstrates the huge potential of matrix-iteration-based algorithms for designing more efficient GPU algorithms in the **flow community**. Hope that our additional experiments will enhance your confidence and recognition of our work.
> ### Reference
> [1] Parallel network simplex algorithm for the minimum cost flow problem. Concurrency and Computation: Practice and Experience, 2022
> [2] Efficient implementations of minimum-cost flow algorithms. arxiv 2012

---

> > ### Comment · Reviewer_vSLu · 2024-11-28
> >
> > Thank you for the further experimental comparisons. I have raised my score.

---

> ### Author Response · Authors · 2024-11-28
>
> May we ask if we have addressed your concern? We do hope we could resolve your any potential questions!

---

### Author Response · Authors · 2024-11-22

## Dear Area Chairs and Reviewers,

In terms of the efficiency of the algorithm, the reviewers deem our work efficient: '**Efficient | GPU-friendly**' (vSLu | 6h6u). Regarding the writing of the paper, the reviewers agree with our writing: '**Well written**' (42rw, T4Wu). In terms of experimental setup, most reviewers consider our work to provide reliable validation: '**Encouraging empirical results**' (6h6u), '**The numerical proposal is sound.**' (T4Wu).

Regarding the contribution of the algorithm, the reviewers agree that it: 'is good for solving large flow problems more **efficiently** than the traditional method.' (vSLu), 'new **interesting** method that can leverage modern GPU infrastructure.' (42rw), and 'more **general** graphs than the bipartite graph.' (42rw).


In the author rebuttal period, we make every effort to address reviewers’ concerns and provide additional experimental results.

At last, we sincerely thank the reviewer and AC for investing their valuable time and efforts in this paper.

---

### Meta-Review · Area_Chair_bemS · 2024-12-20

**Metareview:**

This paper introduces Optimal Flow Transport as an extension of vanilla Optimal Transport to address flow balance constraints in general graph structures, moving beyond the marginal constraints of bipartite graphs used in traditional OT. By incorporating entropic regularization and virtual flows, the authors propose the GPU-friendly, matrix-based OFT-Sinkhorn algorithm to tackle challenges such as isolated nodes and the lack of an alternating iterative structure. They further extend this approach by introducing capacity constraints on nodes and edges, transforming the OFT problem into a capacity-constrained minimum-cost flow (MCF) problem. Empirical evaluations on diverse datasets demonstrate the computational efficiency and scalability of their algorithms, highlighting their ability to approximate optimal solutions effectively while ensuring global convergence.

**Additional Comments On Reviewer Discussion:**

Reviewers express concerns about the novelty of the idea and the strengths of the numerical results. Additionally, the work lacks clear practical applications for machine learning practitioners, making it less impactful for broader audiences. In response, the authors performed considerably more numerical experiments during the rebuttal phase.

---

### Decision · Program_Chairs · 2025-01-22

Accept (Poster)